# Wyckoff Transformer: Generation of Symmetric Crystals

**Nikita Kazeev** [1]  **Wei Nong** [2]  **Ignat Romanov** [3]  **Ruiming Zhu** [2]  **Andrey Ustyuzhanin** [4 5 1]  **Shuya Yamazaki** [2]
**Kedar Hippalgaonkar** [1 2 6]

## Abstract

Crystal symmetry plays a fundamental role in determining its physical, chemical, and electronic properties such as electrical and thermal conductivity, optical and polarization behavior, and mechanical strength. Almost all known crystalline materials have internal symmetry. However, this is often inadequately addressed by existing generative models, making the consistent generation of stable and symmetrically valid crystal structures a significant challenge. We introduce WyFormer, a generative model that directly tackles this by formally conditioning on space group symmetry. It achieves this by using Wyckoff positions as the basis for an elegant, compressed, and discrete structure representation. To model the distribution, we develop a permutation-invariant autoregressive model based on the Transformer encoder and an absence of positional encoding. Extensive experimentation demonstrates WyFormer's compelling combination of attributes: it achieves best-in-class symmetry-conditioned generation, incorporates a physics-motivated inductive bias, produces structures with competitive stability, predicts material properties with competitive accuracy even without atomic coordinates, and exhibits unparalleled inference speed. https://github.com/SymmetryAdvantage/WyckoffTransformer

## 1. Introduction

Discovery of materials with desirable properties is the cornerstone of civilization – from the stone age to the bronze age and now in the silicon age, the ability to wield materials with different properties and function has transformed society (Pyzer-Knapp et al., 2022). However, for the most part, the search of new materials as well as new functionalities, has proceeded through a traditional route of trial-and-error, also called the Edisonian approach (Wang et al., 2024). The space of all possible combinations of atoms forming periodic structures is intractably large, Cao et al. (2024) gauge it at $10^{160}$. It is not possible to fully screen this space or even to enumerate it. Materials that exist under realistic conditions, however, occupy a small part of this set of possibilities (Curtarolo et al., 2013). It consists of the energetically-favored combinations of atoms that are held together through covalent, ionic, metallic and other chemical bonding. A generative model that outputs novel a priori stable materials will speed up automated material design by orders of magnitude.

### 1.1. Space groups and Wyckoff positions

A crystal structure can be represented by lattice vectors and atomic basis. The lattice provides a periodic geometric framework in three-dimensional space, defined by the lattice matrix $\mathbf{L} = [\mathbf{l}_1, \mathbf{l}_2, \mathbf{l}_3] \in \mathbb{R}^{3 \times 3}$, with a basis of an atom (or group of atoms) that occupy any lattice point. The atomic positions in real space are hence given by $\mathbf{X} = [\mathbf{x}_1, \mathbf{x}_2, \ldots, \mathbf{x}_N] \in \mathbb{R}^{3 \times N}$, where $N$ is the number of atoms in the unit cell. These positions can also be expressed in fractional coordinates as $\mathbf{F} = [\mathbf{f}_1, \mathbf{f}_2, \ldots, \mathbf{f}_N] \in [0, 1)^{3 \times N}$, related to real-space coordinates by $\mathbf{F} = \mathbf{L}^{-1}\mathbf{X}$, ensuring atomic positions remain consistent within the periodic lattice. The periodic arrangement can be further constrained by the space group $G$, a finite set of symmetry operations $g \in G$ defined as $g \cdot \mathbf{X} = R\mathbf{X} + \mathbf{t}$, where $R \in O(3)$ is a $3 \times 3$ orthogonal transformation matrix representing rotations, reflections, combinations thereof, and $\mathbf{t} \in \mathbb{R}^3$ is a $3 \times 1$ translation vector. These symmetry operations collectively form the 230 distinct space groups, which comprehensively classify all possible crystal symmetries in three dimensions (Fedorow, 1892; Hahn et al., 1983). Each space group defines the allowable positions for atoms within the

---

[1]Institute for Functional Intelligent Materials University of Singapore, Block S9, Level 9, 4 Science Drive 2, Singapore 117544 [2]School of Materials Science and Engineering, Nanyang Technological University, Singapore 639798 [3]HSE University, Myasnitskaya Ulitsa, 20, Moscow, Russia, 101000 [4]Constructor University, Bremen, Campus Ring 1, 28759, Germany [5]Constructor Knowledge Labs, Bremen, Campus Ring 1, 28759, Germany [6]Institute of Materials Research and Engineering, Agency for Science Technology and Research, 2 Fusionopolis Way, Singapore, 138634. Correspondence to: Nikita Kazeev <kazeevn@gmail.com>, Kedar Hippalgaonkar <kedar@ntu.edu.sg>.

*Proceedings of the 42nd International Conference on Machine Learning*, Vancouver, Canada. PMLR 267, 2025. Copyright 2025 by the author(s).

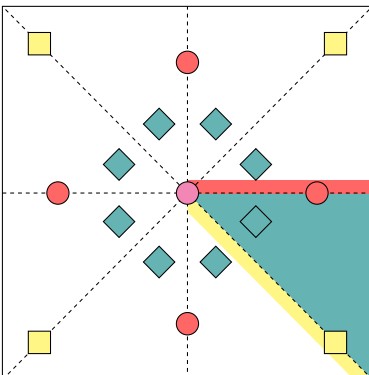

*Figure 1.* A toy 2D crystal (Goodall et al., 2020). It contains 4 mirror lines, and one rotation center. There are four Wyckoff positions, illustrated by shading. Magenta is the Wyckoff position that is invariant under all the transformations, it only contains a single point; red and yellow lie on the mirror lines, and teal is only invariant under the identity transformation and occupies the rest of the space. Markers of the corresponding colors show one of the possible locations of an atom belonging to the corresponding Wyckoff position.

unit cell. Every periodic crystal possesses at least the simplest level of symmetry, P1, which consists only of translational symmetry. Most known crystals have additional internal symmetry, see Figure 2. This is not merely a mathematical observation; optical, electrical, magnetic, structural and other properties are determined by symmetry, as shown by Malgrange et al. (2014); Yang et al. (2005), as well as our results in Section 3.2.

Within a given space group $G$, a subgroup forms the site symmetry, referring to the set of symmetry operations $G_i = \{g \in G \mid g \cdot \mathbf{f}_i \sim \mathbf{f}_i\} \subseteq G$ that leave a specific point in the crystal invariant. These operations describe the local symmetrical environment, such as mirrors, screw axes, or inversions centered on a given region. Atoms located at the representative fractional coordinates $\mathbf{f}_i$ generate equivalent positions $\{R_s \mathbf{f}_i + \mathbf{t}_s\}_{s=1}^{n_s}$, where $n_s$ is the multiplicity of the symmetry-equivalent position. Higher site symmetry is in regions where multiple symmetry elements intersect, while those with lower site symmetry include only one symmetry operation. Taking space group 225 Fm-3m as an example, F represents a face-centered lattice, site symmetry subgroup m-3m represents a highly symmetric environment at the center of a cubic unit cell, where multiple symmetry elements intersect, including mirror planes and a 3-fold rotoinversion axis (Hahn et al., 1983). In contrast, another lower site symmetry subgroup .3m corresponds to a less symmetric environment with only a 3-fold rotation axis and a mirror plane.

These site symmetry points, classified by their symmetry properties, are grouped into Wyckoff positions (WPs)

(Wyckoff, 1922). Mathematically, a WP encompasses all points whose site symmetry groups are conjugate subgroups of the full space group (Kantorovich, 2004). An illustration of WPs is present in Figure 1. Two different WPs in the same space group can share the same site symmetry. This is called symmetry equivalence and occurs when the Wyckoff positions can be mapped onto each other using higher-order symmetry operations. Continuing with the Fm-3m space group example, Wyckoff positions 4a (0,0,0) and 4b (½,½,½) appear distinct under conventional symmetry, but a lattice center translation reveals their higher–order symmetry equivalence (see Figure 3). Such a transformation is a coset representative of the affine normalizer, which introduces symmetry operations beyond the space group's symmetry operations, $G$. The Euclidean normalizer is defined as the largest symmetry group preserving $G$, but allowing additional transformations like centering translations or scaling, mapping Wyckoff positions onto each other in a higher-symmetry framework, forming the basis for enumeration and augmentation in the next sections. We further explore this idea in (Yamazaki et al., 2025).

WPs for a given space group are enumerated by Latin letters, typically in order of decreasing site symmetry. Each WP has a defined multiplicity, which represents the number of equivalent atomic positions in the unit cell related by the symmetry operations of that space group. For example WP 2a has the highest site symmetry and multiplicity 2. The number of distinct WPs in a space group is finite, ranging from a single WP in the simplest symmetry group P1 to as many as 27 in the most complex space groups. Wyckoff positions can represent 1D lines, 2D planes, or open 3D regions within the unit cell. These fundamental concepts — lattice, atomic basis, space groups, site symmetry, and Wyckoff positions — define a framework to unequivocally describe crystal structures, which is the foundation to our representation. See also Appendix A for an illustration.

## 1.2. Our contribution

1. Representing a crystal as an unordered set of tokens fused from the chemical elements and Wyckoff positions; Section 2.1.

2. Encoding Wyckoff positions using their universally defined symmetry point groups and symmetry operations descriptors based on spherical harmonics; Section 2.1.

3. Wyckoff Transformer architecture and training protocol that combine autoregressive probability factorization with permutation invariance; Section 2.3.

4. Model invariance with respect to the arbitrary choice of the coset representative of the space group Euclidean normalizer; Sections 2.1, 2.3.

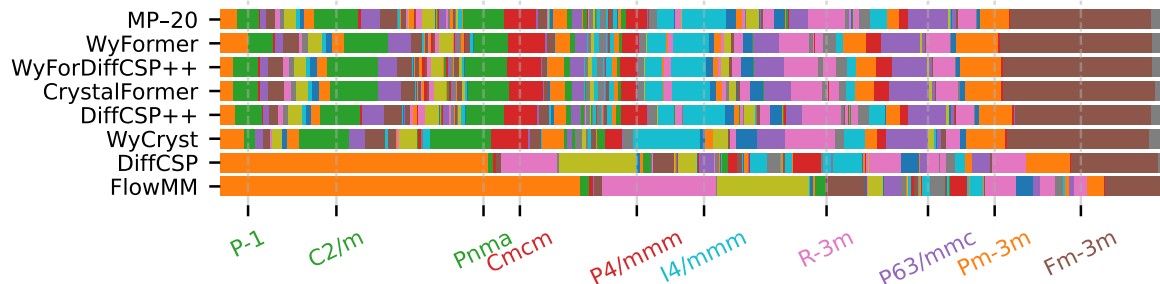

*Figure 2.* Distribution of space groups in MP-20 dataset (Xie et al., 2021) and generated samples. 10 space groups most frequent in MP-20 are labeled, 98% of MP-20 structures belong to symmetry groups other than P1. Plot design by Levy et al. (2024). The comparison of the distribution of generated samples' space groups to the ground truth distribution is presented in Table 1, column Space Group $\chi^2$.

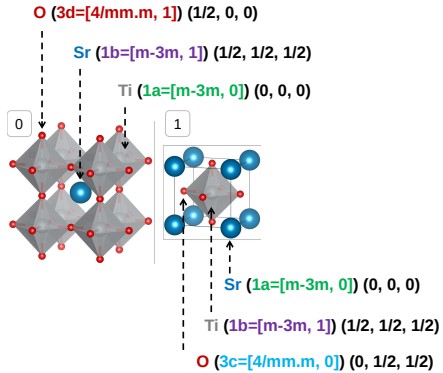

*Figure 3.* Two equivalent Wyckoff representations of SrTiO$_3$ mp-4651, depending on the lattice center choice:
[Ti, (m−3m, 0)], [Sr, (m−3m, 1)], [O, (4/mm.m, 1)]
[Ti, (m−3m, 1)], [Sr, (m−3m, 0)], [O, (4/mm.m, 0)]

5. Empirically, our model outperforms baseline methods in generating novel, symmetric, diverse materials conditioned on space group symmetry; Section 3.2.

6. Despite not using the information about atom coordinates, our model achieves property prediction performance competitive with the machine learning models that use the full structure; Section 3.2.

### 1.3. Related work

**Crystal generation** is a burgeoning field, with most state-of-the-art models using a differentiable non-invertible SO(3) invariant representation constructed from atom coordinates, such as a graph neural network. Then they use diffusion or flow matching to solve the generation problem (Jiao et al., 2024a;b; Cao et al., 2024; Yang et al., 2023; Zeni et al., 2025; Xie et al., 2021; Klipfel et al., 2023; Luo et al., 2024; Sinha et al., 2024). Our approach uses discrete Wyckoff space and fast autoregressive sampling, compared to gradual refinement in the aforementioned works. WyFormer complements them naturally by providing symmetry constraints and/or initial structure approximation — the synergy with the most

suitable partner, DiffCSP++, we evaluate thoroughly.

**Wyckoff positions and machine learning.** The concept of Wyckoff positions was originally published more than a 100 years ago (Wyckoff, 1922). Given the elegance of the representation, naturally, in modern times WPs have found their way into machine learning. The main limiting factor in their adoption was the ability of machine learning algorithms to handle discrete structured data which is formed by WPs. WP-based representation was used for property prediction (Goodall et al., 2020; Jain & Bligaard, 2018; Möller et al., 2018; Goodall et al., 2022), and recently in generative models. Our work is inspired by Zhu et al. (2024), the first such model. It uses a VAE over one-hot-encoded information about WPs, as opposed to our Transformer encoder, which is a generally superior architecture for categorical data. AI4Science et al. (2023) use GFlowNet (Bengio et al., 2023) to sample space group and chemical composition, but not the full Wyckoff representation. A concurrent preprint (Cao et al., 2024) independently explores a Transformer-based approach similar to ours; another concurrent work (Levy et al., 2024) uses diffusion over Wyckoff position site symmetry, fractional coordinates, and lattice parameters.

The main difference between our and most other approaches, that are based on Wyckoff positions is that they use Wyckoff letters as the representation. Wyckoff letter definitions depend on the space group, unlike site symmetry, leading to data fragmentation. Levy et al. (2024) also use WP site symmetry, with one-hot-encoding of symmetry operations per axis to represent it; Goodall et al. (2022) use the sum of one-hot-encodings of sites to represent a WP; we treat site symmetry as a categorical variable and use learnable embeddings. Zhu et al. (2024); Cao et al. (2024) don't take into account dependency of the Wyckoff letters on the arbitrary choice of the coset representative of the space group Euclidean normalizer. Finally, Cao et al. (2024) use positional encoding to establish the relationship between the chemical elements and Wyckoff positions they occupy, while we combine them in one token.

Spherical harmonics are widely used to build a fixed-length descriptor for spatial relationships (Bartók et al., 2013).

## 2. Wyckoff Transformer (WyFormer)

### 2.1. Tokenization

Our work is based on the inductive bias that for stable materials space group symmetry and Wyckoff sites almost completely define the structure – more than 98% of the materials in MP-20 (Xie et al., 2021) and MPTS-52 (Baird et al., 2024) datasets, which together contain almost all experimentally stable structures from the Materials Project (Jain et al., 2013), have unique Wyckoff representations. Therefore, it is safe to assume that for almost any Wyckoff representation there is either none, or just one stable material conforming to it. Symmetry captured by this discrete part is sufficient to determine properties of a material, such as piezoelectricity via non-centrosymmetry; direct/indirect band gap via positions of the valence/conduction bands in the Brillouin Zone, while the fractional coordinates can be linked to the magnitude of that property. We additionally prove this assumption by predicting various material properties; see Section 3.2. Given a Wyckoff representation, coordinates can be determined as discussed in Section 2.4.

We represent each structure as a set of tokens, as shown in Figure4. The first token contains the space group; the rest are divided into groups of three tokens, each representing a specific WP. The first token in each group is responsible for the type of atom that occupies the position following the site symmetry, while the last token is for the so-called *enumeration*. Several WPs can have the same site symmetry. To differentiate those WPs we enumerate them separately within each space group and site symmetry according to the conventional WP order (Aroyo et al., 2006). For example, in space group 225 present in Figure4 WP 4a is encoded as (m-3m, 0), 4b as (m-3m, 1), and 8c as (-43m, 0). A more comprehensive example can be found in Appendix S. The purpose of this encoding is to take advantage of the fact that, unlike Wyckoff letters, site symmetry definition is universal across different space groups. An ablation study comparing our representation with Wyckoff letters is in Appendix N.

Such an encoding has an additional advantage. For a given crystal, the conventional unit cell can sometimes be chosen in several equivalent ways, which changes the Wyckoff positions (see Figure 3) corresponding to each atom, but not their site symmetries. We collect all the arbitrariness in one variable, which leaves the rest of the representation strictly invariant to that choice.

Formally, we define Wyckoff representation of a structure as $R = (G, \mathbf{E}, \mathbf{W})$, where $G$ is the space group, $W = [w_1, \ldots, w_m]$ are the Wyckoff positions with $w_i = (s_i, n_i)$,

where $s_i$ is the site symmetry, and $n_i$ is the enumeration, and $E = [e_1, \ldots, e_m]$ are the chemical elements occupying them. Spglib (Atsushi Togo & Tanaka, 2024) provides us a mapping $\rho$ from crystal $C = (L, E, F)$ to $R$, which is used to preprocess the training dataset. The problem solved by Wyckoff Transformer is sampling the distribution $P(R| \exists C : \rho(C)$ is stable), which is enabled by learning the following probabilities: $p(e_i|G, E_{i-1}, S_{i-1}, N_{i-1})$, $p(s_i|E_i, S_{i-1}, N_{i-1})$, and $p(n_i|E_I, S_I, N_{i-1})$, where $E_{i-1} = [e_1, \ldots, e_{i-1}]$, $S_{i-1} = [s_1, \ldots, s_{i-1}]$, $N_{i-1} = [n_1, \ldots, n_{i-1}]$.

#### 2.1.1. SPHERICAL HARMONICS

*Enumerations* are defined by an arbitrary convention, in this respect they are no better than Wyckoff letters. We address this with a representation that is defined consistently across space groups. Consider a Wyckoff position consisting of a set of $k$ symmetry operations $\{R_i \boldsymbol{x} + \boldsymbol{t}_i, i = 1 \ldots k\}$. We apply these operations to points $\boldsymbol{x}_1 = [0, 0, 0]$ and $\boldsymbol{x}_2 = [1, 1, 1]$ obtaining two matrices $W^{(1)}$ and $W^{(2)}$: $W_i^{(j)} = R_i \boldsymbol{x}_j + \boldsymbol{t}_i \boldsymbol{x}_j$. Finally, we convolve the transformed coordinates with spherical harmonics:

$$\boldsymbol{\phi}_i^{(j)} = \arctan([W^{(j)}]_i^2, W^{(j)}]_i^1); \boldsymbol{\theta}_i^{(j)} = \arccos([W^{(j)}]_i^3)$$

$$\boldsymbol{h}^{(j)} = \sum_{i=1}^k |W_i^{(j)}|[Y_n^0(\boldsymbol{\theta}_i^{(j)}, \boldsymbol{\phi}_i^{(j)}), \ldots, Y_n^n(\boldsymbol{\theta}_i^{(j)}, \boldsymbol{\phi}_i^{(j)})]/k,$$

where $n$ is the degree of spherical harmonics, a parameter, and the resulting complex vectors $\boldsymbol{h}^{(1)}$ and $\boldsymbol{h}^{(2)}$ each have $n + 1$ dimensions. $n = 2$ is enough to disambiguate all Wyckoff positions with the same site symmetry belonging to the same space groups; $n = 1$ is not. Finally, we obtain the final $2n + 2$ dimensional descriptor $\boldsymbol{s}$ by concatenation: $\boldsymbol{s} = \Re(\boldsymbol{h}^{(1)} \oplus \boldsymbol{h}^{(2)}) \oplus \Im(\boldsymbol{h}^{(1)} \oplus \boldsymbol{h}^{(2)})$. The harmonic representation is not directly invertible; in the main section of the paper, we only use it for property prediction, which results in a slight performance increase, as shown in Appendix O. A way to adapt the harmonics-based representation for structure generation is discussed in Appendix P.

### 2.2. Model architecture

Elements, site symmetries, and *enumerations* are each embedded with a simple lookup table with trainable weights, the embeddings are concatenated. Then we apply a linear layer to provide each head of the multihead attention with information from all three parts of a token.

Since our model is conditioned on space group, preventing data fragmentation is of utmost importance. To this end, the space group is not encoded just as a categorical variable. Building upon (AI4Science et al., 2023) and similarly to Levy et al. (2024) we use pyXtal to get one-hot-encoded $15 \times 10$ matrix that represents symmetry elements on each

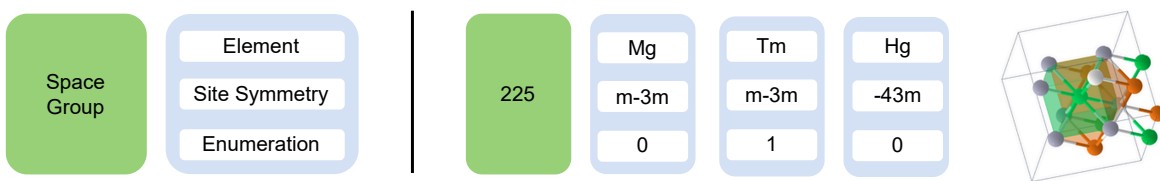

*Figure 4.* An example of structure tokenization, TmMgHg$_2$ mp-865981

axis for each space group, flatten it, discard the positions that do not vary across the dataset and use the resulting vector as the space group embedding. Then we apply a linear layer, so the representation becomes learnable — but still transferable between space groups.

Token sequences are used as input for a Transformer encoder (Vaswani, 2017; Devlin, 2018). Wyckoff representation is permutation-invariant, so is Transformer; we do not use positional encoding, making the model formally permutation-invariant with respect to the input.

**De novo generation** We use *enumerations* representation. We additionally add a STOP token to each structure. To represent states where some parts of token are known and others are not, we replace those values with MASK. We also add a fully-connected neural network for each part of the token that we want to predict, three in total. To get the prediction, we take the output of Transformer encoder on the token containing MASK value(s), concatenate it with a one-hot vector encoding presence in the input sequence of each possible value for this token part, and use it as the input for the corresponding fully-connected network.

**Property prediction** We take the Transformer encoder outputs tokens, excluding the token corresponding to the space group, compute a weighted average with weights being equal to the multiplicities of WPs, and use the result as input for a fully-connected neural network that outputs a scalar predicted value.

### 2.3. Training

Following the approach of Wang et al. (2023); Abramson et al. (2024), we use a simple architecture and do not strictly enforce invariance with respect to the choice of the equivalent Wyckoff representations, but rather leave it as a training goal by picking a randomly selected equivalent representation at every training epoch. It is especially viable because of the low number of variants; in MP-20 dataset for 96% structures there are less than 10.

The experimental results we present were obtained by training separate models for property prediction and de novo generation. A single model to do both is possible, we leave it for the future work.

**De novo generation** The training pipeline and architecture are shown in Figure 5. We train the model to predict next part of a token in a cascade fashion: first the chemical element conditioned on the previous tokens, then site symmetry conditioned on the previous tokens and the element and, finally, *enumeration* conditioned on the previous tokens, the element and the site symmetry. On each training iteration we randomly sample known sequence length and the part of the cascade to predict; place MASK tokens as necessary, input the known parts of the sequences into the model, compute cross-entropy loss between the predicted scores and the target.

Unlike Transformer itself, auto-regressive generation is not permutation-invariant. The number of WPs is small, the average in MP-20 is just 3.0; this again allows us to train the model to be invariant with augmentation by shuffling the order of every Wyckoff representation at every training epoch. Moreover, we use multi-class loss when training to predict the first cascade part, chemical element, further reducing learning complexity.

On MP-20 the model is trained for $9 \times 10^5$ epochs using SGD optimizer without batching; due to the efficiency of the representation gradient backpropagation for the entire dataset fits into GPU memory. We use the loss on the validation dataset for early stopping, learning rate scheduling, and manual hyperparameter tuning.

**Property prediction** The model is trained using MSE loss with batch size 500, and Adam optimizer. For both MP-20 and AFLOW training takes around 5k epochs.

Hyperparameters are available in Appendix L.

### 2.4. Structure generation

We generate crystals conditioned on space group number which is sampled from the combination of training and validation datasets, as illustrated in Figure 7. Wyckoff representation is then autoregressively sampled using WyFormer. We use two ways to generate the final crystal structure conditioned on the representation, the details are described in Appendix C. They both start with randomly sampling a structure conditioned on the Wyckoff representation with pyXtal (Fredericks et al., 2021). Then it's relaxed with CrySPR (Nong et al., 2024) and CHGNet (Deng et al., 2023)

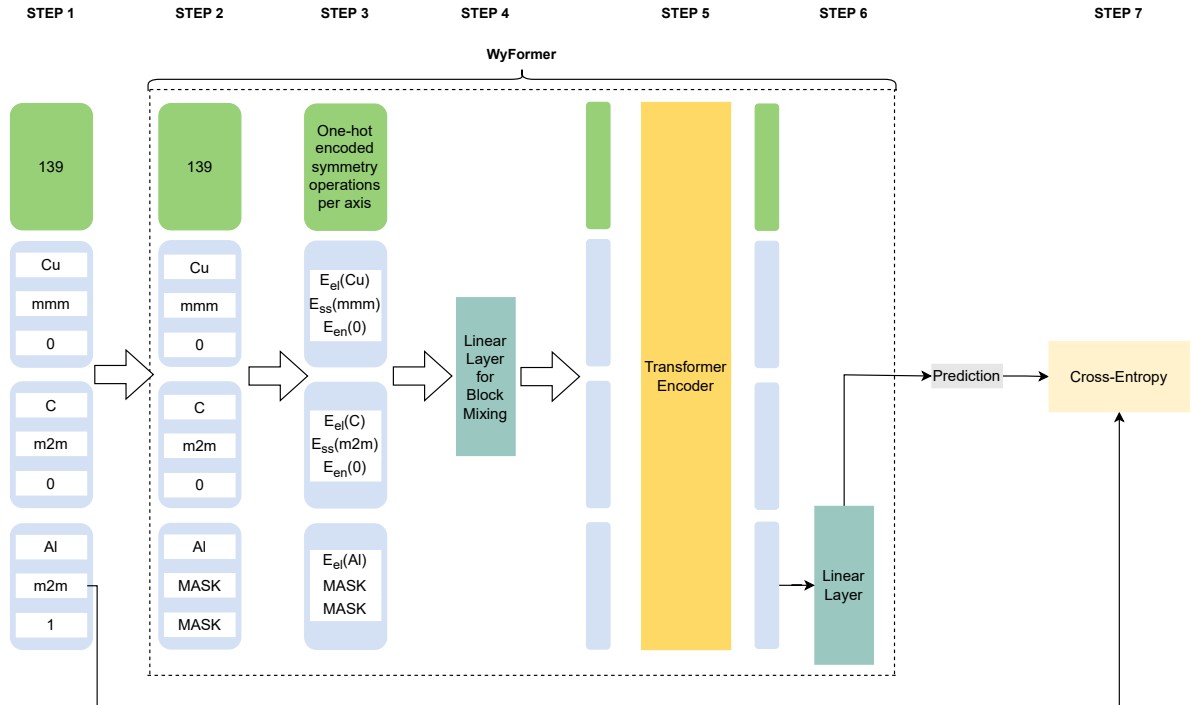

*Figure 5.* Model training pipeline. (1) The crystal is converted into a token sequence where the first token is the space group number and then token triplets in the order atom, site, symmetry and enumeration. Then the triplets are randomly shuffled. (2) Randomly sample the number of fully known Wyckoff positions and the part of the next triplet to be predicted; mask unknown tokens, remove unknown Wyckoff positions. (3) Embed the tokens using simple lookup tables; for each Wyckoff positions concatenate tokens corresponding to it in the embedding dimension. (4) A linear layer mixes the features to provide homogeneous input to multiple attention heads. (5) The sequence is passed through the Transformer Encoder. (6) An MLP is applied to the last token of the output sequence. (7) The loss is cross entropy of the prediction and the true value of the token being predicted.

or DiffCSP++ (Jiao et al., 2024b).

# 3. Experimental Evaluation

## 3.1. De novo generation

### 3.1.1. DATASETS

We use MP-20 (Xie et al., 2021), which contains almost all experimentally stable materials in Materials Project (Jain et al., 2013) with a maximum of 20 atoms per unit cell, within 0.08 eV/atom of the convex hull, and formation energy smaller than 2 eV/atom, 45 229 structures in total, split 60/20/20 into train, validation and test parts. Additionally, we train and evaluate WyFormer on MPTS-52 (Baird et al., 2024), a more challenging subset of Materials Projects containing materials with up to 52 atoms per unit cell.

### 3.1.2. METRICS

**Structure property similarity metrics** Coverage and Property EMD (Wasserstein) distance, have been proposed as a low-cost proxy metric for de novo structure generation by

Xie et al. (2021) and then followed by most of the subsequent work.

**Validity** Xie et al. (2021) proposed verifying crystal feasibility according to two criteria:

*Structural* validity means that no two atoms are closer than 0.5Å. All structures in MP-20 and almost all structures produced by state-of-the-art models fulfill it.

*Compositional* validity means having neutral charge (Davies et al., 2019). Only 90% of MP-20 structures pass this test meaning that nonconforming structures are physically possible if somewhat rare.

**Novelty and uniqueness** The purpose of de novo generation is to obtain new materials. Generated materials that already exist in the training dataset increase the model performance according to structure stability and similarity metrics, but such structures are useless for material design and just increase the gap between the proxy metrics and the model fitness for its purpose. Therefore we exclude generated materials that are not novel and unique from metric computation. On a deeper level, generative models for materials

*Table 1.* Evaluation. Symmetry metrics are computed only using novel structurally valid examples. Note that the 1000 and 105-example metrics are computed using MP-20 train and validation as reference datasets for novelty, while the 10 000-example S.U.N. only uses MP-20 train to remain compatible with the reported values. **Bold** indicates the values within $p = 0.1$ statistical significance threshold from the best. Values marked by $^*$ were computed by Miller et al. (2024), the rest by us; see note H.1 for an important caveat; in short, the values in (brackets) are less accurate, but are compatible with each other.

| Method/Metric | Novel Unique Templates (#) $\uparrow$ | P1 (%) ref = 1.7 | Space Group $\chi^2 \downarrow$ | S.U.N. % $\uparrow$ $E_{hull} < 80$ meV | S.S.U.N. % $\uparrow$ $E_{hull} < 80$ meV | S.U.N. % $\uparrow$ $E_{hull} < 0$ meV |
|---|---|---|---|---|---|---|
| *Sample size* | 1000 | 1000 | 1000 | 105 | 105 | 10 000 |
| *Relaxation* | CHGNet | CHGNet | CHGNet | DFT | DFT | DFT |
| WyFormerCHGNet | **180** | 3.24 | 0.223 | **23.1** | **22.3** | – |
| WyFormerDiffCSP++ | **186** | **1.46** | **0.212** | 22.2 | 21.1 | **3.83** (4.14) |
| DiffCSP++ | *10* | 2.57 | 0.255 | 14.4 | **14.4** | – |
| CrystalFormer | 74 | **0.91** | 0.276 | **20.1** | 20.1 | – |
| SymmCD | 101 | **2.35** | 0.24 | **20.7** | 20.7 | – |
| WyCryst | **165** | 4.79 | 0.710 | 5.5 | 5.5 | – |
| DiffCSP | 76 | 36.57 | 7.989 | **22.2** | 20.6 | – (3.34$^*$) |
| FlowMM | 51 | 44.27 | 12.423 | **17.8** | **16.9** | – (2.34$^*$) |
| WyFormer *MPTS–52* | 386 | 0 | 0.225 | – | – | – |

are subject to exploration/exploitation trade-off: the more physically similar are the sampled materials to the training dataset, the more likely they are stable and distributed similar to the data, but the less useful they are for the purpose of material design. From a purely machine learning point of view, novelty percentage serves a proxy metric for overfitting.

**Stability** determines whether the material, in fact, exists under normal conditions. It is estimated by computing energy above convex hull, and comparing it to a threshold. Materials Project is the source of the reference structures for the hull. The details are in Appendix G.

**S.U.N.** (Zeni et al., 2025) combines the above into the fraction of stable unique novel structures.

**Symmetry** of the structures has paramount physical importance. Controlling symmetries also leads to control over physical, electronic, and mechanical behavior, which is desirable in property-directed inverse design of materials. For example, in electronic materials, higher symmetry can improve carrier mobility and uniformity in electronic band structure, enhancing performance in applications such as semiconductors or optoelectronics. Furthermore, high-symmetry structures often exhibit isotropic properties, meaning their behaviors are the same in all directions, making them more versatile for industrial use. We use four metrics for evaluating the ability of the generative models to reproduce the symmetry present in the data and, ultimately, in nature:

**– P1** is the percentage of the structures that have symmetry group P1. In MP-20 the corresponding number is just 1.7%.

We argue that presence of symmetry is good proxy value for structure feasibility that is difficult to capture in standard DFT computations, and would require finite-temperature calculations and/or improved methodologies.

**– Novel Unique Templates** is the number of the novel unique element-agnostic Wyckoff representations (Section 2.1) in the generated sample. Element-agnostic means that we remove the chemical element, while retaining the symmetry information. For example, for the TmMgHg$_2$ in Figure 4, it will be `(X, (m-3m, 0))`, `(X, (m-3m, 1))`, `(X, (-43m, 0))` and its equivalent. An important difference between our work and (Levy et al., 2024) is that we take into account equivalence of Wyckoff representations. The metric provides a lower limit on overfitting and physically meaningful sample novelty: if two materials have different symmetry templates, their physical properties will be different, while the inverse is not always true. It serves as an addition to the strict structure novelty, which provides the upper bound. Finally, the ability of a model to generate new templates allows it generate more structures before starting to repeat itself, as we demonstrate in Appendix J.

**– Space Group $\chi^2$** is the $\chi^2$ statistic of difference of the frequencies of space groups between the generated and test datasets.

**– S.S.U.N.** is the percentage of the structures that are symmetric (space group not P1), stable, unique and novel.

### 3.1.3. METHODOLOGY

**WyFormer** was trained using MP-20 dataset following the original train/test/validation split. We sampled

$10^4$ Wyckoff representations, then obtained structures using CrySPR+CHGNet (WyFormerCHGNet) and DiffCSP++ (WyFormerDiffCSP++) approaches described in Section 3.1.3.

**WyCryst** (Zhu et al., 2024) only supports a limited number of unique elements per structure, therefore we trained it on a subset of MP-20 containing only binary and ternary compounds, 35 575 in total. An evaluation of WyFormer trained on the same dataset is present in Appendix K. As WyCryst also produces Wyckoff representations, and not structures, the same CrySPR+CHGNet procedure was used to obtain them.

**CrystalFormer** (Cao et al., 2024) code and weights published by the authors were used by us to produce the sample, conditioned on the space groups sampled from MP-20.

**DiffCSP** (Jiao et al., 2024a), **DiffCSP++** (Jiao et al., 2024b), and **SymmCD** (Levy et al., 2024) samples were provided by the authors. The DiffCSP++ sampling process is conditioned on Wyckoff templates from the training dataset, which includes the space group.

For each model a data sample containing 1000 structures was relaxed using CHGNet. The generated samples were filtered for uniqueness, more than 99.5% of structures for every method passed the filtering. We computed for DFT for 105 novel structures for each method; detailed description of the settings is available in Appendix H.

Additionally, we computed DFT for 10 000 structures from WyFormer, and compared S.U.N. values to the values reported by Miller et al. (2024).

### 3.1.4. DE NOVO STRUCTURE GENERATION RESULTS

Evaluation results are present in Tables 1 and 2; a sample of generated structures is illustrated in Figure 9.

**WyFormer** achieves 24% higher S.U.N. on the 10 000-structure sample compared to the best available baseline; best template novelty, fraction of asymmetric structures and space group distribution reproduction. On the 105-structure sample, the difference WyFormer, CrystalFormer, DiffCSP, FlowMM, and SymmCD the difference between S.U.N. and S.S.U.N. values is not statistically significant.

**DiffCSP++** has lower stability, despite using a priori valid structure templates from the data. As we show in Appendix J, the lack of template novelty limits the diversity, and the model starts to repeat itself. DiffCSP++ oversamples the structures with the large number of unique elements, WyFormer matches the distribution most closely, as depicted in Figure 10.

**CrystalFormer** has lower novelty, which means that the model has been overfitted, and the structures are more similar to the training dataset. It also produces a sizable fraction of a priori structurally invalid crystals.

**WyCryst** suffers from even lower novelty, stability and distribution similarity metrics.

**DiffCSP and FlowMM** can not be conditioned on the symmetry group, and produce a large fraction of unrealistic asymmetric structures.

**SymmCD** is a concurrent work based on similar principles, and achieves similar performance, except for a lesser number of Novel Unique Templates.

**On MPTS-52**, as expected, WyFormer shows higher novelty as well as template novelty. In terms of distribution similarity metrics WyFormer performs largely similarly on MP-20 and MPTS-52. We used CHGNet to predict formation energies estimate S.S.U.N.: 24.4% on MPTS-52, compared to 35.2% on MP-20. This reflects the increased difficulty, and shows that WyFormer is still very much capable of generating stable structures in this setting.

### 3.2. Material property prediction

MP-20 dataset contains two properties: formation energy and band gap, which we predict using WyFormer. The results are shown in Table 3. WyFormer achieves competitive results with the models that use full structures.

We also utilize the AFLOW database (Curtarolo et al., 2012), which contains 4905 compounds spanning a diverse range of chemistries and crystal structures. We predict four properties: thermal conductivity, Debye temperature, bulk modulus, and shear modulus. The data are divided into training, validation, and test sets using a 60/20/20 split. The results are presented in Table 4; WyFormer demonstrated superior performance in predicting thermal conductivity. For the remaining three properties, the model's performance is comparable to the baselines.

From this we argue that the symmetries and composition of a crystal alone already carry a considerable amount of information about its properties. This is especially true for band gap, where Brillouin zones are defined by symmetry, and thermal conductivity, which is a non-equilibrium phonon transport property conditioned on underlying symmetry of the structure; according to the first order approximation kinetic theory, higher symmetry crystals typically have higher thermal conductivity due to (1) higher group velocities and (2) longer scattering times due to lower anharmonicity (Newnham, 2004; Yang et al., 2021).

## 4. Conclusions and Limitations

$E_{\text{hull}}$ determined from formation energy as a proxy for stability is commonly used, but is imperfect, as it doesn't take

*Table 2.* Evaluation of the methods according to validity and property distribution metrics. Structures were relaxed with CHGNet. Following the reasoning in Section 3.1.2, we apply filtering by novelty and structural validity, and do not discard structures based on compositional validity. An evaluation following the protocol proposed by Xie et al. (2021) is available in Appendix I.

| Method | Novelty (%) ↑ | Validity (%) ↑ | | Coverage (%) ↑ | | Property EMD ↓ | | |
|---|---|---|---|---|---|---|---|---|
| | | Struct. | Comp. | COV-R | COV-P | $\rho$ | $E$ | $N_{\text{elem}}$ |
| WyFormerCHGNet | 90.00 | 99.56 | 80.44 | 98.67 | 96.72 | 0.74 | 0.053 | **0.097** |
| WyFormerDiffCSP++ | 89.50 | 99.66 | 80.34 | 99.22 | 96.79 | 0.67 | 0.050 | 0.098 |
| DiffCSP++ | 89.69 | **100.00** | **85.04** | 99.33 | 95.80 | **0.15** | 0.036 | 0.504 |
| CrystalFormer | *76.92* | 86.84 | 82.37 | **99.87** | 95.13 | 0.52 | 0.100 | 0.163 |
| SymmCD | 88.77 | 95.82 | 84.88 | 99.55 | 94.66 | 0.62 | 0.102 | 0.525 |
| WyCryst | *52.62* | 99.81 | 75.53 | 98.85 | 87.10 | 0.96 | 0.113 | 0.286 |
| DiffCSP | **90.06** | **100.00** | 80.94 | 99.55 | 96.21 | 0.82 | 0.052 | 0.294 |
| FlowMM | 89.44 | **100.00** | 81.93 | 99.67 | **99.64** | 0.49 | **0.036** | 0.131 |
| WyFormer MPTS–52 | 98.7% | 99.3% | 76.7% | – | – | 0.698 | 0.108 | 0.228 |

*Table 3.* One-shot energy and band gap prediction. We computed CHGNet energy predictions on the MP-20 dataset, the rest of the baseline values are from (Lin et al., 2023); The MP-20 test set is a part of CHGNet training set. Xie & Grossman (2018); Jha et al. (2019) report the error between DFT-computed and experimental results $\approx 0.08$ eV for energy, and $\approx 0.6$ eV for band gap.

| Method | Energy meV | Band gap meV | Train | Test |
|---|---|---|---|---|
| CGCNN | 31 | 292 | | |
| SchNet | 33 | 345 | | |
| MEGNet | 30 | 307 | Materials | |
| GATGNN | 33 | 280 | Project | |
| ALIGNN | 22 | 218 | 2018.6.1 | |
| Matformer | 21 | 211 | | |
| PotNet | **19** | **204** | | |
| CHGNet | 34 | – | MPTrj | MP-20 |
| WyFormer | 25 | 234 | MP-20 | |

*Table 4.* MAE values for AFLOW dataset; baseline values are by Wang et al. (2021).

| Method | Thermal conductivity | Debye temperature | Bulk modulus | Shear modulus |
|---|---|---|---|---|
| Roost | 2.70 | 37.17 | 8.82 | 9.98 |
| CrabNet | 2.32 | **33.46** | **8.69** | **9.08** |
| HotCrab | 2.25 | 35.76 | 9.10 | 9.43 |
| ElemNet | 3.32 | 45.72 | 12.12 | 13.32 |
| RF | 2.66 | 36.48 | 11.91 | 10.09 |
| WyFormer | **2.20** | 36.36 | 9.63 | 10.14 |

An important part of the future work is Crystal Structure Prediction (CSP). Unlike the models that work with atoms and coordinates, it is hard to ensure that WyFormer output strictly conforms to a given stoichiometry. But we can add the stoichiometry as a generation condition, like space group. Then, as as we show in Appendix 6, WyFormer is four orders of magnitude faster than other CSP solutions, which allows to simply use rejection sampling.

In conclusion, we demonstrate that WyFormer represents a novel advancement in generation of realistic symmetric crystals by leveraging Wyckoff positions to encode material symmetries. WyFormer achieves a higher degree of structure diversity compared to baselines by encoding the discrete symmetries of space groups without relying on atomic coordinates. This unique tokenization of symmetry elements enables the model to explore a reduced, yet highly representative space of possible configurations, resulting in more stable and purportedly synthesizable crystals. The model respects the inherent symmetry of crystalline materials, outperforms existing models in generating both novel and physically meaningful structures. These innovations underscore the method's potential in accelerating material discovery while maintaining accuracy in predicting key properties like formation energy and band gap.

into account configurational and vibrational entropic contributions, and hull determination relies on already known structures. Moreover, our results, along with Miller et al. (2024) show that generated structures with space symmetry group P1 are consistently found stable at a much higher rate than they occur in nature. There are two logical explanations: either DiffCSP and FlowMM have, in passing, discovered a new class of asymmetric materials – or our stability estimation methodology is systematically flawed. In our biased opinion the latter is much more likely.

Novelty and diversity evaluation is a crucial and open question. A model can generate structures that are similar to the ones in the training dataset, and are valid, but not very useful for new material design. Counting complete duplicates is a step in the right direction, but doesn't measure substantial sample diversity (Hicks et al., 2021).

# Acknowledgements

We thank Lei Wang for insights on symmetry-conditioned generation; Andrey Okhotin for insights on permutation invariance and the 10k CHGNet computation; Benjamin Miller for a discussion of the evaluation metrics; Rui Jiao and Daniel Levy for providing data samples.

This research/project is supported by the Ministry of Education, Singapore, under its Research Centre of Excellence award to the Institute for Functional Intelligent Materials (I-FIM, project No. EDUNC-33-18-279-V12). This research/project is supported by the National Research Foundation, Singapore under its AI Singapore Programme (AISG Award No: AISG3-RP-2022-028) and from the MAT-GDT Program at A*STAR via the AME Programmatic Fund by the Agency for Science, Technology and Research under Grant No. M24N4b0034. The computational work for this article was performed on resources at the National Supercomputing Centre of Singapore (NSCC). Computational work involved in this research work is partially supported by NUS IT's Research Computing group. The research used computational resources provided by Constructor Tech. This research was supported in part through computational resources of HPC facilities at HSE University.

# Impact Statement

This paper presents work whose goal is to advance the field of Machine Learning. There are many potential societal consequences of our work, none which we feel must be specifically highlighted here.

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

## A. Wyckoff representation with fractional coordinates

A crystal can be represented as a space group, a set of WPs and chemical elements occupying them, the fractional coordinates of the WP degrees of freedom, and free lattice parameters. Such representation reduces the number of parameters by an order of magnitude without information loss. For example, see Figure 6.

```
Group: I4/mmm (139)
```
Lattice: $a = b = \mathbf{8.9013}, \ c = \mathbf{5.1991}, \ \alpha = 90.0, \ \beta = 90.0, \ \gamma = 90.0$
```
Wyckoff sites:
Nd @ [ 0.0000  0.0000  0.0000], WP [2a] Site [4/m2/m2/m]
Al @ [ 0.2788  0.5000  0.0000], WP [8j] Site [mm2.]
Al @ [ 0.6511  0.0000  0.0000], WP [8i] Site [mm2.]
Cu @ [ 0.2500  0.2500  0.2500], WP [8f] Site [..2/m]
```

*Figure 6.* Wyckoff representation of $Nd(Al_2Cu)_4$ (mp-974729), variable parameters in **bold**. If represented as a point cloud, the structure has 13[atoms] $\times$ 3[coordinates] + 6[lattice] = 42 parameters; if represented using WPs, it has just 4 continuous parameters (WPs 8i and 8j each have a free parameter, and the tetragonal lattice has two), and 5 discrete parameters (space group number, and WPs for each atom).

## B. WyFormer Description

Structure generation is shown in Figure 7 and described in Algorithm 1; training in Algorithm 2; model itself in Algorithm 3.

## C. Structure generation details

The process of obtaining crystal structures from Wyckoff representations using PyXtal (Fredericks et al., 2021) begins by specifying a space group and defining WPs. PyXtal allows users to input atomic species, stoichiometry, and symmetry preferences. Based on these parameters, PyXtal generates a random crystal structure that respects the symmetry requirements of the space group. Once the initial structure is generated, we then perform energy relaxation using CHGNet. CHGNet is a neural network-based model designed to predict atomic forces and energies, significantly speeding up calculations that would traditionally require density functional theory (DFT). We repeat the process for six random initializations and pick the structure with the lowest energy. Energy distribution among the initializations is presented in Figure 8. Energy relaxation involves optimizing the atomic positions to reach a minimum energy configuration, which represents the most stable form of the material. CHGNet, trained on vast DFT datasets, can efficiently relax crystal structures by adjusting atomic positions to reduce the total energy. This approach ensures that the final structure is not only symmetrical but also physically realistic in terms of energy stability.

For the 2nd structure generation method, DiffCSP++ is a diffusion-based crystal structure prediction model that focuses on generating purportedly stable crystal structures by sampling from an energy landscape in a physically consistent manner. DiffCSP++ generation also starts with PyXtal sampling.

## D. Training computational requirements

Our tests were done on a single NVIDIA RTX 6000 Ada, 24 CPU cores and MP-20 dataset. The results are present in Figure5. We have also tried batched training for next token prediction training, but it just becomes slower without improved

*Table 5.* WyFormer training resources requirements on the MP-20 dataset.

| Prediction target | Time | Batch size | Number of epochs | GPU memory, MiB | GPU load |
|---|---|---|---|---|---|
| Next token | 11h | 27136 | 900k | 2000 | 50% |
| Formation energy | 26m | 1000 | 5.2k | 1700 | 45% |
| Band gap | 10m | 1000 | 3k | 1700 | 25% |

quality. The reason might be that we choose a random known sequence length and part of the token, token permutation, and

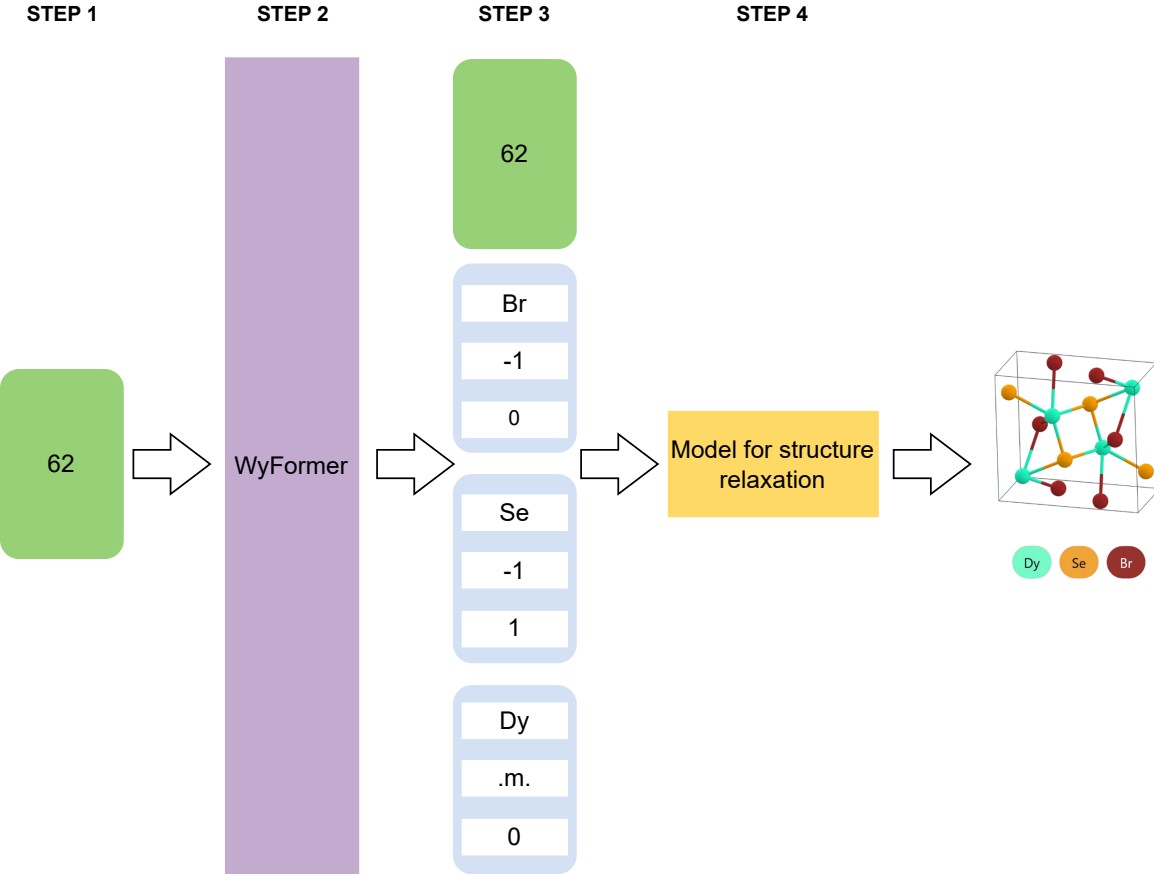

*Figure 7.* High-level flowchart of structure generation with WyFormer. In step 1 space group which is sampled from the training data distribution and used as the initial token for WyFormer; in step 2 WyFormer autoregressively generates tokens; in step 3 the Wyckoff representation is converted to JSON and stored. Finally, in step 4, the Wyckoff representation is passed to DiffCSP++/CrySPR for structure generation as described in Section 2.4.

---

**Algorithm 1** Generation of Crystal Structure using Wyckoff Transformer Model

---

1: Load Trained Wyckoff Transformer Model
2: Select or sample a *space_group*
3: Initialize *sequence* = [*space_group*]
4: *loopShouldEnd* ← **false**
5: *current_token_count* ← length(*sequence*)
6: **repeat**
7:    **if** *current_token_count* ≥ *max_length* **then**
8:       *loopShouldEnd* ← **true**
9:    **end if**
10:    **if not** *loopShouldEnd* **then**
11:       Predict *element* using Model(*sequence*)
12:       **if** *element* == *STOP* **then**
13:          *loopShouldEnd* ← **true**
14:       **else**
15:          **append** *element* to *sequence*
16:          *current_token_count* ← *current_token_count* + 1
17:          **if** *current_token_count* ≥ *max_length* **then**
18:             *loopShouldEnd* ← **true**
19:          **end if**
20:       **end if**
21:    **end if**
22:    **if not** *loopShouldEnd* **then**
23:       Predict *site_symmetry* using Model(*sequence*)
24:       **if** *site_symmetry* == *STOP* [Less likely, but possible] **then**
25:          *loopShouldEnd* ← **true**
26:       **else**
27:          **append** *site_symmetry* to *sequence*
28:          *current_token_count* ← *current_token_count* + 1
29:          **if** *current_token_count* ≥ *max_length* **then**
30:             *loopShouldEnd* ← **true**
31:          **end if**
32:       **end if**
33:    **end if**
34:    **if not** *loopShouldEnd* **then**
35:       Predict *enumeration* using Model(*sequence*)
36:       **if** *enumeration* == *STOP* [Less likely, but possible] **then**
37:          *loopShouldEnd* ← **true**
38:       **else**
39:          **append** *enumeration* to *sequence*
40:          *current_token_count* ← *current_token_count* + 1
41:          **if** *current_token_count* ≥ *max_length* **then**
42:             *loopShouldEnd* ← **true**
43:          **end if**
44:       **end if**
45:    **end if**
46: **until** *loopShouldEnd* = **true**
47: Convert generated *sequence* of (*element*, *site_symmetry*, *enumeration*) tokens into a list of {*element*, Wyckoff position letter} pairs for the chosen *space_group*.
48: Use pyXtal library with the Wyckoff representation to create an initial 3D crystal structure.
49: Relax the structure a MLIP (CHGNet, etc.), DiffCSP++, or DFT
50: **return** the crystal structure.

---

---

**Algorithm 2** Wyckoff Transformer Training Algorithm

---

**Require:** Training dataset $D_{\text{train}}$ (crystal structures represented as sequences of tokens: space_group + list of [element, site_symmetry, enumeration])

1: Initialize Wyckoff Transformer Model $M$ with random weights
2: Initialize Optimizer $O$ (e.g., SGD)
3: **for** epoch = 1 **to** MaxEpochs **do**
4:     **for** each crystal structure sequence $S$ in $D_{\text{train}}$ **do**
5:         $S_{\text{aug}} \leftarrow S$
6:         Randomly shuffle the order of [element, site_symmetry, enumeration] tokens within $S_{\text{aug}}$ {A}ugmentation for permutation invariance
7:         Randomly choose one of the equivalent Wyckoff representations for $S_{\text{aug}}$
8:         $\text{pos}_{\text{target}} \leftarrow$ Randomly pick a position in $S_{\text{aug}}$ to predict
9:         $\text{part}_{\text{target}} \leftarrow$ Randomly pick which part of the token (element, site_symmetry, or enumeration) to predict at $\text{pos}_{\text{target}}$
10:       Replace $\text{part}_{\text{target}}$ at $\text{pos}_{\text{target}}$ in $S_{\text{aug}}$ with a MASK token
11:       Also mask any subsequent parts of the token at $\text{pos}_{\text{target}}$, remove the tokens after it
12:       $P_{\text{pred}} \leftarrow M(S_{\text{aug}})$ {Forward Pass: Model predicts the masked part}
13:       $V_{\text{actual}} \leftarrow$ the true value of $\text{part}_{\text{target}}$ at $\text{pos}_{\text{target}}$ in the original $S_{\text{aug}}$
14:       $L \leftarrow \text{CrossEntropyLoss}(P_{\text{pred}}, V_{\text{actual}})$ {Use multi-class variant if $\text{part}_{\text{target}}$ is element}
15:       Calculate gradients $\nabla L$ based on the loss $L$ w.r.t. $M$'s parameters
16:       Update $M$'s weights using $O(\nabla L)$
17:     **end for**
18:     *Optional:* Validate model $M$'s performance on a separate validation dataset $D_{\text{val}}$ periodically.
19:     *Optional:* Adjust learning rate or implement early stopping based on validation performance.
20: **end for**
21: **return** Wyckoff Transformer Model $M$.

---

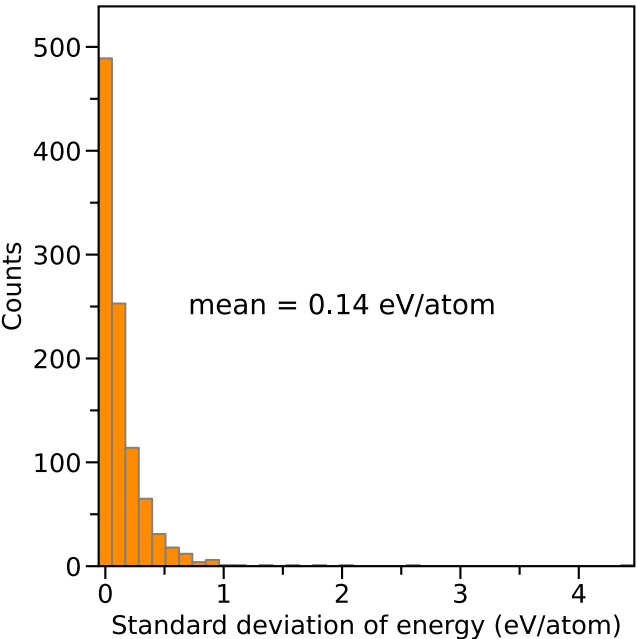

*Figure 8.* Distribution of CHGNet-predicted energy standard deviation across six random pyXtal initializations for 1000 Wyckoff representations.

---

**Algorithm 3** Model Forward Pass

---

1: **Define** element_embedding_layer (lookup table)
2: **Define** site_symmetry_embedding_layer (lookup table)
3: **Define** enumeration_embedding_layer (lookup table)
4: **Define** space_group_embedding_layer (special encoding + linear layer)
5: **Define** embedding_mixer (linear layer) {Mixes concatenated [element, site_symmetry, enumeration] embeddings}
6: **Define** transformer_encoder_block (standard Transformer Encoder layers, NO positional encoding)
7: **Define** element_prediction_head (Fully-connected Neural Network)
8: **Define** site_symmetry_prediction_head (Fully-connected Neural Network)
9: **Define** enumeration_prediction_head (Fully-connected Neural Network)
10:
11: **Function** Model_Forward (space_group_token, sequence_of_wyckoff_tokens)
12: space_group_embedding ← space_group_embedding_layer(space_group_token)
13: wyckoff_embeddings_list ← []
14: **for** each token in sequence_of_wyckoff_tokens **do**
15:     element_emb ← element_embedding_layer(token.element)
16:     site_sym_emb ← site_symmetry_embedding_layer(token.site_symmetry)
17:     enum_emb ← enumeration_embedding_layer(token.enumeration)
18:     concatenated_emb ← **concaternate**(element_emb, site_sym_emb, enum_emb)
19:     mixed_wyckoff_emb ← embedding_mixer(concatenated_emb)
20:     **append** mixed_wyckoff_emb **to** wyckoff_embeddings_list
21: **end for**
22: full_sequence_embeddings ← **concatenate**(space_group_embedding, wyckoff_embeddings_list)
23: transformer_output_sequence ← transformer_encoder_block(full_sequence_embeddings)
24: target_scores ← transformer_output_sequence.last {the masked token is the last one}
25: *Optional*: target_embedding ← **concatenate**(target_embedding, presence_vector)
26: **if** predicting_element **then**
27:     predicted_probabilities ← element_prediction_head(target_embedding)
28: **else if** predicting_site_symmetry **then**
29:     predicted_probabilities ← site_symmetry_prediction_head(target_embedding)
30: **else if** predicting_enumeration **then**
31:     predicted_probabilities ← enumeration_prediction_head(target_embedding)
32: **end if**
33: **return** predicted_probabilities

---

*enumerations* variant on every batch; this can help to avoid a sharp minimum even when gradients are computed over the whole dataset.

For comparison, training DiffCSP++ took 19.5 hours and 32000 MiB of GPU memory.

## E. Inference speed

We conducted experiments on a machine with NVIDIA RTX 6000 Ada and 24 physical CPU cores. For baselines, we used source code, model hyperparameters and weights published by the authors. Assuming that the downstream costs of structure relaxation by DFT or machine-learning interaction potential are fixed, the inference cost per S.U.N. structure is present in the Figure6.

*Table 6.* Inference time per S.U.N. structure. When a GPU is running, it also occupies a CPU core, which is taken into account. S.U.N. rates are measured according to DFT stability estimation. CHGNet is not used anywhere, for WyFormerRaw we sample a structure with pyXtal and use it directly as an input for DFT.

| Method | S.U.N. | GPU ms per | | CPU s per | |
|---|---|---|---|---|---|
| | (%) | structure | S.U.N. | structure | S.U.N. |
| WyFormerRaw | 4.8 | **0.05** | **1.0** | **0.105** | **2.2** |
| WyForDiffCSP++ | 12.8 | 840 | 5957 | 0.940 | 6.7 |
| DiffCSP | 19.7 | 360 | 1731 | 0.360 | 1.73 |
| DiffCSP++ | 7.6 | 1250 | 14705 | 1.35 | 15.9 |

Generating a batch of Wyckoff representations takes 25 seconds, of which 5 seconds are spent generating PyTorch tensors, and 20 seconds on decoding them into Python dictionaries containing Wyckoff representations. The latter part has not been optimized. In total, generation takes 0.05 GPU ms and 4.8 CPU ms per structure.

Obtaining unrelaxed structures using pyXtal takes 100 CPU ms / structure.

Relaxing the structure is the most expensive step. DiffCSP++ takes 14 minutes to produce 1000 structures at 840 GPU ms / structure. Note that we modified the code to remove the inference of atom types, so it runs faster compared to the original version. CHGNet: 112 GPU s / structure for MP-20 on NVIDIA A40

Baselines

- DiffCSP: the authors don't report speed. On our machine, generating 10000 structures on GPU took 1 hour, at 360 GPU ms per structure.

- DiffCSP++: the authors don't report speed. On our machine, generating 27135 structures took 6 hours, at 1.25 CPU+GPU seconds per structure

- CrystalFormer; Cao et al. (2024): "It takes 520 seconds to generate a batch size 13,000 crystal samples on a single A100 GPU", which translates to a generation speed of 40 ms per sample.

- FlowMM: The authors also do not publish inference time or model weights. They claim to be 3x faster than DiffCSP in terms of integration steps.

- WyCryst; Zhu et al. (2024): "Latent space sampling 1 CPU second/2000 structures; PyXtal generation 2 CPU core seconds/structure"

## F. Plots

Figure 10 contains the number of unique elements per structure for MP-20 and novel generated structures.

## G. Energy above hull calculations

For CHGNet, to obtain the $E_{\text{hull}}$, we firstly constructed the reference convex hull data by querying all 153235 structures from the Materials Project (MP); then, using the `pymatgen.analysis.phase_diagram` sub-module the $E_{\text{hull}}$ for

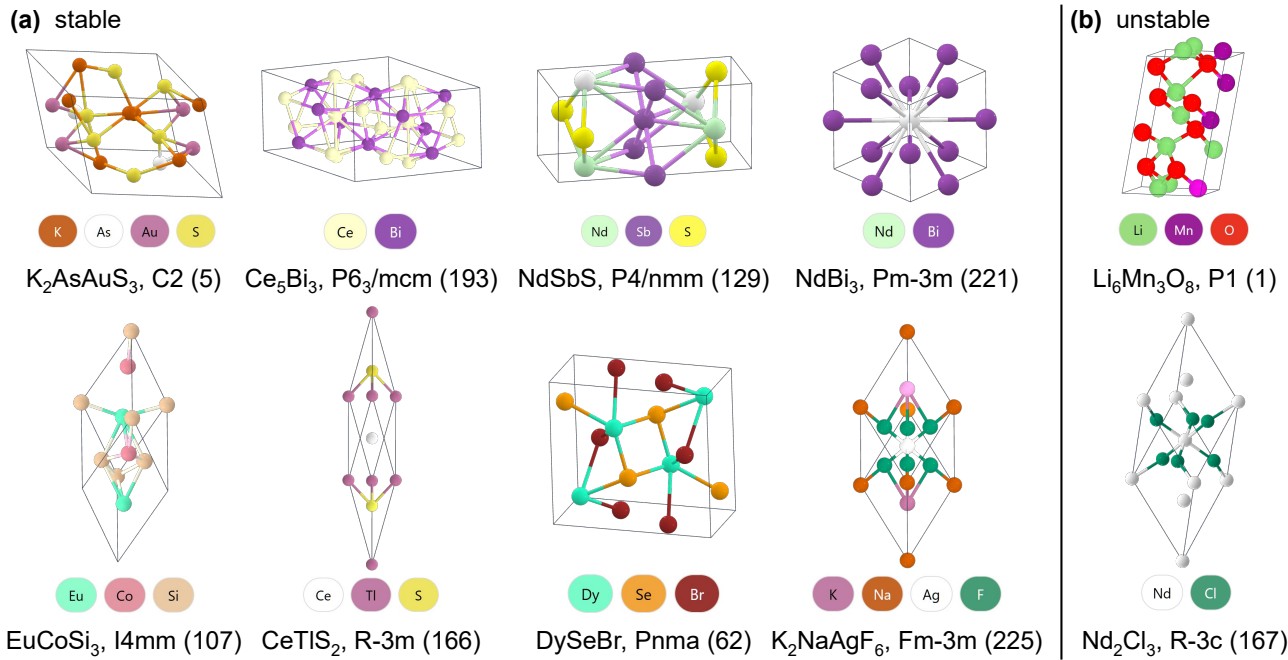

**(a)** stable

K$_2$AsAuS$_3$, C2 (5)   Ce$_5$Bi$_3$, P6$_3$/mcm (193)   NdSbS, P4/nmm (129)   NdBi$_3$, Pm-3m (221)

**(b)** unstable

Li$_6$Mn$_3$O$_8$, P1 (1)

EuCoSi$_3$, I4mm (107)   CeTlS$_2$, R-3m (166)   DySeBr, Pnma (62)   K$_2$NaAgF$_6$, Fm-3m (225)   Nd$_2$Cl$_3$, R-3c (167)

*Figure 9.* 10 structures generated from WyFormerDiffCSP++ and presented without additional relaxation. The labels contain the chemical formula, followed by the space group symbol in the short Hermann-Mauguin notation, and space group number. To the left 8 structures were randomly chosen from 15 stable structures as validated by DFT calculations, to the right 2 from unstable structures. The solid box lines represent the primitive cell.

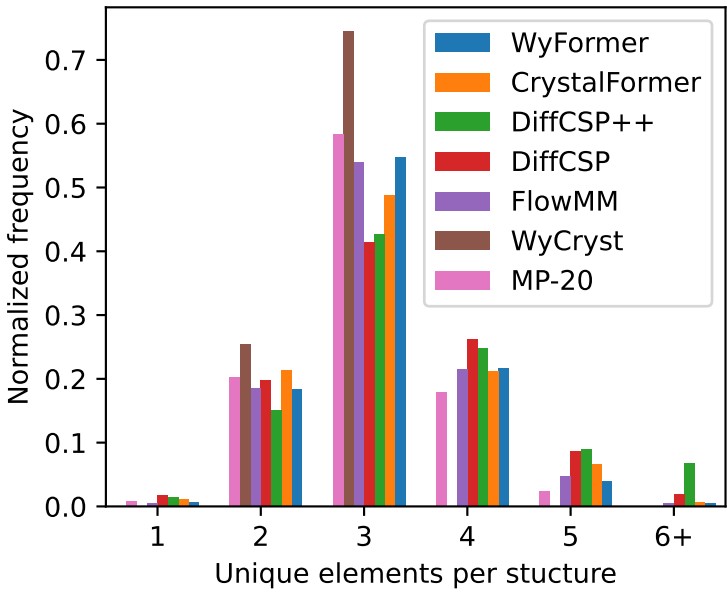

*Figure 10.* Distribution of the number of unique elements per structure for MP-20 and novel generated structures.

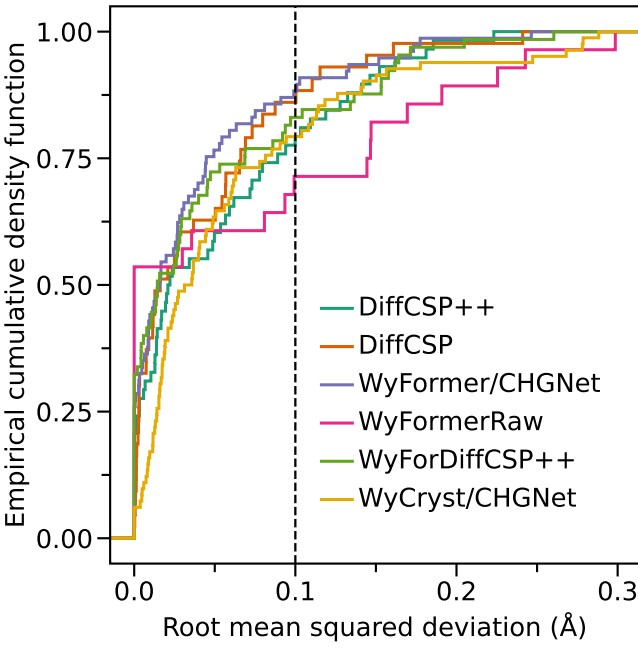

*Figure 11.* The empirical cumulative density function (ECDF) for root mean squared deviation (RMSD) of DFT-unrelaxed structures from DFT-relaxed counterparts. RMSD is calculated using `pymatgen.analysis.StructureMatcher`, in which only the RMSD of matched structure pairs is reported. WyFormer/CHGNet and WyCryst/CHGNet refer to the models that use CHGNet-relaxed structures as inputs for DFT relaxations, while WyFormerRaw refers to WyFormer directly using pyxtal-generated unrelaxed structures (Section C).

each entry of generated structure was computed by referencing to the MP convex hull, $E_{hull} = \max\{\Delta E_i\}$, where $\Delta E_i$ is the decomposition energy of any possible path for a structure decomposing into the reference convex hull. For the DFT data we used the MP convex hull `2023-02-07-ppd-mp.pkl.gz` distributed by `matbench-discovery` (Riebesell et al., 2023) was used as the reference hull.

## H. DFT details

We use DFT settings from Materials Project https://docs.materialsproject.org/methodology/materials-methodology/calculation-details/gga+u-calculations/parameters-and-convergence for structure relaxation and energy computation. In particular, we do GGA and GGA+U calculations with `atomate2.vasp.flows.mp. MPGGADoubleRelaxStaticMaker` (Ganose et al., 2025), which in turn relies on `pymatgen.io.vasp.sets.MPRelaxSet` and `pymatgen.io.vasp.sets.MPStaticSet` (Ong et al., 2013). Computations themselves were done with VASP (Kresse & Furthmüller, 1996) version 5.4.4. with the plane-wave basis set (Kresse & Furthmüller, 1996). The electron-ion interaction is described by the projector augmented wave (PAW) pseudo-potentials (Kresse & Joubert, 1999). The exchange-correlation of valence electrons is treated with the Perdew-Burke-Ernzerhof (PBE) functional within the generalized gradient approximation (GGA) (Perdew et al., 1996).

For a small fraction $(1-15\%)$ of the generated structures, the DFT failed to converge. We consider such structures to be unstable for the purposes of S.U.N. computation. The effect is especially strong for CrystalFormer, as 13% of the structures it generates are structurally invalid, that is have atoms closer than 0.5 Å.

The 105-sample relaxations used structures as produced by the ML models. For the 10 000-sample run we used CHGNet pre-relaxation to speed up the computations.

The raw total energies computed by DFT were corrected with `MaterialsProject2020Compatibility` before putting into the `PhaseDiagram` to obtain the DFT $E_{hull}$.

## H.1. DFT setting difference between Materials Project and (Miller et al., 2024)

The settings used by Miller et al. (2024) are not entirely the same as the ones used by the Materials Project, us, and Zeni et al. (2025). Both in the paper (section A.7.) and in the code `https://github.com/facebookresearch/flowmm/blob/6a96aec3b6eba89f6fa07436f0c8837979abb285/scripts_analysis/dft_create_inputs.py#L43`, Miller et al. (2024) refer to using `pymatgen.io.vasp.sets.MPRelaxSet`, and doing so only once. The Materials Project procedure consists of relaxing all cell and atomic positions two times in consecutive runs, followed by a more precise static calculation `https://docs.materialsproject.org/methodology/materials-methodology/calculation-details`.

To estimate the effect and make a direct comparison, in Figure 1 we report the S.U.N. obtained from structures relaxed with single `MPRelaxSet` in (brackets) and our more accurate `MPGGADoubleRelaxStaticMaker` result without.

## I. Legacy metrics

For completeness sake, in Figure 7 we present the metrics computed following the protocol set up by Xie et al. (2021). We would like to again reiterate the issues with it. Firstly, the metrics are negatively correlated with structure novelty, the raison d'être for material generative models. Secondly, filtering by charge neutrality aka compositional validity means discarding viable structures.

*Table 7.* Method comparison according the protocol set up by Xie et al. (2021). COV-P depends on the generated sample size, so to compute it we uniformly subsample 1k structures.

(a) Directly using structures produced by the methods, without additional relaxation. Note that CHGNet is an integral part of generating structures with Wyckoff Transformer and WyCryst, so it's used.

| Method | Validity (%) ↑ | | Coverage (%) ↑ | | Property EMD ↓ | | |
|---|---|---|---|---|---|---|---|
| | Struct. | Comp. | COV-R | COV-P | $\rho$ | $E$ | $N_{elem}$ |
| WyckoffTransformer | 99.60 | 81.40 | 98.77 | 95.94 | 0.39 | 0.078 | 0.081 |
| WyFormerDiffCSP++ | 99.80 | 81.40 | 99.51 | 95.81 | 0.36 | 0.083 | **0.079** |
| DiffCSP++ | 99.94 | **85.13** | 99.67 | 95.71 | 0.31 | **0.069** | 0.399 |
| CrystalFormer | 93.39 | 84.98 | 99.62 | 94.56 | 0.19 | 0.208 | 0.128 |
| SymmCD | **100.00** | 86.27 | 99.50 | 94.82 | **0.06** | 0.160 | 0.402 |
| WyCryst | 99.90 | 82.09 | 99.63 | 96.16 | 0.44 | 0.330 | 0.322 |
| DiffCSP | **100.00** | 83.20 | **99.82** | 96.84 | 0.35 | 0.095 | 0.347 |
| FlowMM | 96.87 | 83.11 | 99.73 | 95.59 | **0.12** | 0.073 | 0.094 |

(b) All structures have been relaxed with CHGNet. Note that for some models we didn't compute CHNet relaxation for all the structures, so the sample size is smaller.

| Method | Validity (%) ↑ | | Coverage (%) ↑ | | Property EMD ↓ | | |
|---|---|---|---|---|---|---|---|
| | Struct. | Comp. | COV-R | COV-P | $\rho$ | $E$ | $N_{elem}$ |
| WyckoffTransformer | 99.60 | 81.40 | 98.77 | 95.94 | 0.39 | 0.078 | 0.081 |
| WyTransDiffCSP++ | 99.70 | 81.40 | 99.26 | 95.85 | 0.33 | 0.070 | **0.078** |
| DiffCSP++ | **100.00** | **85.80** | 99.42 | 95.48 | **0.13** | **0.036** | 0.453 |
| CrystalFormer | 89.92 | 84.88 | **99.87** | 95.45 | 0.19 | 0.139 | 0.119 |
| SymmCD | 95.49 | 85.86 | 99.19 | 96.05 | 0.32 | 0.095 | 0.392 |
| WyCryst | 99.90 | 82.09 | 99.63 | 96.16 | 0.44 | 0.330 | 0.322 |
| DiffCSP | **100.00** | 82.50 | 99.64 | 95.18 | 0.46 | 0.075 | 0.321 |
| FlowMM | **100.00** | 82.83 | 99.71 | 95.83 | 0.17 | 0.046 | 0.093 |

## J. Template Novelty and Diversity

To asses the impact of template novelty on the diversity of the generated data can be assessed by evaluating the number of unique structures as the function of the total dataset size. We sampled 118k examples from the model with the lowest template novelty, DiffCSP++, and the highest, WyFormer. We present the number of unique samples as a function of the

generated sample size in Figure 12. DiffCSP++ uniqueness is clearly lower; due to its high inference costs (see Appendix 6), we were unable to prepare a larger sample.

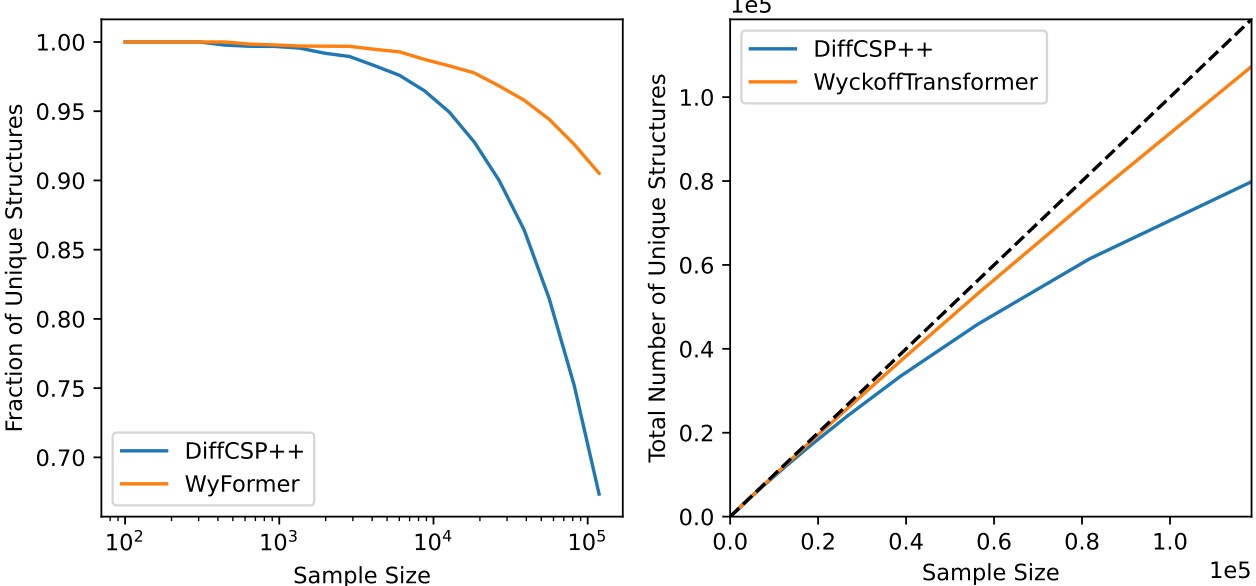

*Figure 12.* Fraction of unique structures and total number of unique structures as a function of sample size. For Wyckoff Transformer we used only the Wyckoff representations for uniqueness assessment, meaning that the uniqueness is likely to be slightly underestimated.

## K. Evaluation on MP-20 binary & ternary

Comparison of WyFormer to WyCryst is presented in tables 8 and 9. Both models were trained on a subset of MP-20 training data containing only binary and ternary structures, and similarly selected subset of MP-20 testing dataset is used as the reference for property distributions. All generated structures were relaxed with CHGNet.

WyFormer outperforms WyCryst across the board. S.U.N. values are close, but this is achieved by WyCryst sacrificing sample diversity and property similarity metrics, with about half of the generated structures already existing in the training dataset.

*Table 8.* Evaluation of the methods according to the symmetry metrics. Aside from Template Novelty, metrics are computed only using novel structurally valid structures. Stability estimated with CHGNet.

| Method | Template Novelty (%) ↑ | P1 (%) ref = 1.7 | Space Group $\chi^2$ ↓ | S.S.U.N. (%) ↑ |
|---|---|---|---|---|
| WyFormer | **25.63** | **1.43** | **0.224** | **37.9** |
| WyCryst | 18.51 | 4.79 | 0.815 | 35.2 |

*Table 9.* Evaluation of the methods according to validity and property distribution metrics. Following the reasoning in Section 3.1.2, we apply filtering by novelty and structural validity, and do not discard structures based on compositional validity. Validity is also computed only for novel structures. Stability estimated with CHGNet.

| Method | Novelty (%) ↑ | Validity (%) ↑ Struct. | Comp. | Coverage (%) ↑ COV-R | COV-P | Property EMD ↓ $\rho$ | $E$ | $N_{elem}$ | S.U.N. (%) ↑ |
|---|---|---|---|---|---|---|---|---|---|
| WyFormer | **91.19** | **99.89** | **77.28** | **98.90** | **96.75** | **0.83** | **0.064** | 0.084 | **38.4** |
| WyCryst | 52.62 | 99.81 | 75.53 | 98.85 | 89.27 | 1.35 | 0.128 | **0.003** | 36.6 |

# L. Hyperparameters

## L.1. Next token prediction MP-20

Model:

- **WP representation:** Site symmetry + enumeration
- **Element embedding size:** 16
- **Site symmetry embedding size:** 16
- **Site enumerations embedding size:** 8
- **Number of fully-connected layers:** 3
- **Number of attention heads:** 4
- **Dimension of feed–forward layers inside Encoder:** 128
- **Dropout inside Encoder:** 0.2
- **Number of Encoder layers:** 3

Optimizer:

- **Loss function:** Cross Entropy, multi-class for element, single-class for other token parts, no averaging
- **Batch size:** 27136 (full MP-20 train)
- **Optimizer:** SGD
- **Initial learning rate:** 0.2
- **Scheduler:** `ReduceLROnPlateau`
- **Scheduler patience:** $2 \times 10^4$ epochs
- **Early stopping patience:** $10^5$ epochs of no improvement in validation loss
- **clip_grad_norm:** max_norm=2

## L.2. Energy prediction MP-20

Model:

- **WP representation:** Site symmetry + harmonics
- **Element embedding size:** 32
- **Site symmetry embedding size:** 64
- **Harmonics vector size:** 12
- **Embedding dropout:** 0.03
- **Number of fully-connected layers:** 3
- **Fully-connected dropout:** 0
- **Number of attention heads:** 4
- **Dimension of feed–forward layers inside Encoder:** 128

- **Dropout inside Encoder:** 0.1
- **Number of Encoder layers:** 4

Optimizer:

- **Loss function:** Mean squared error (MSE), averaged over batch
- **Batch size:** 1000
- **Optimizer:** Adam
- **Initial learning rate:** 0.001
- **Scheduler:** ReduceLROnPlateau
- **Scheduler patience:** 200 epochs
- **Scheduler factor:** 0.5
- **Early stopping patience:** $10^3$ epochs of no improvement in validation loss
- **clip_grad_norm:** max_norm=2

### L.3. Band gap prediction MP-20

Model:

- **WP representation:** Site symmetry + harmonics
- **Element embedding size:** 32
- **Site symmetry embedding size:** 64
- **Harmonics vector size:** 12
- **Embedding dropout:** 0.05
- **Number of fully-connected layers:** 3
- **Fully-connected dropout:** 0.03
- **Number of attention heads:** 4
- **Dimension of feed–forward layers inside Encoder:** 128
- **Dropout inside Encoder:** 0.2
- **Number of Encoder layers:** 1

Optimizer:

- **Loss function:** Mean squared error (MSE), averaged over batch
- **Batch size:** 1000
- **Optimizer:** Adam
- **Initial learning rate:** 0.001
- **Scheduler:** ReduceLROnPlateau
- **Scheduler patience:** 200 epochs
- **Scheduler factor:** 0.5
- **Early stopping patience:** $10^3$ epochs of no improvement in validation loss
- **clip_grad_norm:** max_norm=2

## M. Fine-tuning LLM with Wyckoff representation

To challenge Wyckoff Transformer's architecture, we compared it with pre-trained language models that were used in vanilla mode as well as after fine-tuning, essentially combining approach by Gruver et al. (2024) with Wyckoff representation. We explored two different textual representations of crystals corresponding to a given space group:

- **Naive**, which contains the specifications of atoms at particular symmetry groups encoded by Wyckoff symmetry labels: `Na at a, Na at a, Na at a, Mn at a, Co at a, Ni at a, O at a, O at a, O at a, O at a, O at a, O at a`

- **Augmented**, which contains the specifications of atom types with its' symmetries and site enumerations: `Na @ m @ 0, Na @ m @ 0, Na @ m @ 0, Mn @ m @ 0, Co @ m @ 0, Ni @ m @ 0, O @ m @ 0, O @ m @ 0, O @ m @ 0, O @ m @ 0, O @ m @ 0, O @ m @ 0`, where the set of valid symmetries is: `['2.22', '4/mmm', '1', '-3..', '6mm', 'm-3m', '2', '3mm', '.m', '-6mm2m', '4mm', '.32', '322', '.2/m.', '-1', '.m.', '..m', 'm.2m', '.3m', '3m', 'm2m.', '2mm', '-32/m.', '2..', '..2', '.3.', '2/m', '-43m', '4/mm.m', '.2.', '2/m2/m.', '23.', '222', 'm..', 'mm.', '-3.', 'm-3.', '3.', '4/m..', '.-3m', '2m.', '-32/m', '-42m', 'm.mm', '4..', 'm.m2', '422', '32.', '22.', '-622m2', '3m.', '.-3.', 'mmm..', '222.', 'mm2..', '-4m2', '2/m..', 'mm2', '-3m2/m', '-4m.2', '2mm.', '3..', '-42.m', '..2/m', '4m.m', '-4..', '6/mm2/m', 'm2m', 'm2.', '2.mm', 'mmm.', 'mmm', '32', 'm', '-6..']`

We fine-tuned the OpenAI `chatGPT-4o-mini-2024-07-18` model using different representations and compared it with the vanilla OpenAI `gpt-4o-2024-08-06` model. For each of the cases prompt looked like: `Provide example of a material for spacegroup number X`. The table below contains details of the model training:

| Model | Base Model | Representation | Hyperparameters | Training Time | Inference Time | Number of Parameters |
|---|---|---|---|---|---|---|
| WyLLM-vanilla | gpt-4o-2024-08-06 | Naive | – | – | 74m | $\approx 200B$ |
| WyLLM-naive | gpt-4o-mini-2024-07-18 | Naive | epochs: 1, batch: 24, learning rate multiplier: 1.8 | 51m | 51m | $\approx 8B$ |
| WyLLM-site-symmetry | gpt-4o-mini-2024-07-18 | Site Symmetry | epochs: 1, batch: 24, learning rate multiplier: 1.8 | 95m | 37m | $\approx 8B$ |

*Table 10.* Comparison of different models and their characteristics. Number of parameters is not known exactly and is taken from public sources as an approximate estimation. For reference, WyFormer has 150k parameters.

Both training and inference times were measured using batch job execution on OpenAI's cloud. The fine-tuned model returned a JSON string that was easy to parse, while the vanilla model required additional parsing of its output.

Comparison the WyFormer to WyLLM is present in Figure11. When fine-tuned, an LLM using Wyckoff representations shows similar performance to WyFormer – at a much greater computational cost. Using site symmetries instead of Wyckoff letters doesn't unequivocally increase the LLM performance, a possible explanation is that since this representation is our original proposition, the LLM is less able to take advantage of pre-training that contained letter-based Wyckoff representations. Without fine-tuning, the majority of LLM outputs are formally invalid, and the distribution of the valid ones doesn't match MP-20.

## N. Ablation study: letters vs site symmetries

To evaluate the effect of using a representation based on site symmetry, as opposed on Wyckoff letters, we trained a WyFormer model with the same hyperparameters, but using a Wyckoff letters, and not site symmetry + enumeration representation. The letter-based variant underperforms, as show in Figure12.

Table 11. Comparison for WyFormer to different variant of WyLLM. All structures have been relaxed with DiffCSP++. Sample size is 1000 structures per model. The metrics described in Section 3.1.2. nan is placed where the generated structures contained a rare element that crashed the property computation code. Wyckoff Validity refers to the percentage of the generated outputs that are valid Wyckoff representations. Aside from LLM-specific problems, such as non-existent elements, a Wyckoff representation can be invalid if it places several atoms at Wyckoff position without degrees of freedom, or refers to Wyckoff positions that do not exist in the space group. Stability computed with DFT.

| Method | Novelty | Validity (%) ↑ | | Coverage (%) ↑ | | Property EMD ↓ | | |
|---|---|---|---|---|---|---|---|---|
| | (%) ↑ | Struct. | Comp. | COV-R | COV-P | $\rho$ | $E$ | $N_{\text{elem}}$ |
| WyFormer | 89.50 | 99.66 | 80.34 | 99.22 | **96.79** | 0.67 | **0.050** | 0.098 |
| WyLLM-naive | 94.67 | 99.79 | 82.89 | 98.72 | 94.97 | 0.39 | 0.067 | **0.015** |
| WyLLM-vanilla | **95.59** | 99.82 | **88.75** | 94.46 | 59.67 | 2.23 | 0.234 | 0.253 |
| WyLLM-site-symmetry | 89.58 | **99.89** | 83.89 | **99.44** | 96.32 | **0.29** | nan | 0.039 |

| Method | Wyckoff Validity | Novel Unique | $P1$ (%) | Space Group | S.U.N. | S.S.U.N. |
|---|---|---|---|---|---|---|
| | (%) ↑ | Templates (#) ↑ | ref = 1.7 | $\chi^2$ ↓ | % ↑ | % ↑ |
| WyFormer | **97.8** | 186 | 1.46 | 0.212 | **22.2** | **21.3** |
| WyLLM-naive | 94.9 | **237** | **1.38** | 0.167 | 11.7 | 11.7 |
| WyLLM-vanilla | 28.7 | 87 | 2.03 | 0.621 | | |
| WyLLM-site-symmetry | 89.6 | 191 | 2.24 | **0.158** | | |

| Method/Metric | Novel Unique | P1 (%) | Space Group | S.U.N. | S.S.U.N. |
|---|---|---|---|---|---|
| | Templates (#) ↑ | ref = 1.7 | $\chi^2$ ↓ | % ↑ | % ↑ |
| WyFormerDiffCSP++ | 186 | **1.46** | **0.21** | **22.2** | **21.1** |
| WyFormer-letters-DiffCSP++ | **250** | 1.16 | **0.21** | 16.0 | 16.0 |

Table 12. WyFormer using Wyckoff letters (WyFormer-letters-DiffCSP++) vs WyFormer using site symmetry+enumeration (WyFormerDiffCSP++)

## O. Performance analysis of encoding WPs with spherical harmonics

To assess the impact of spherical harmonics we compare the performance of models with the same set of hyperparameters for the property prediction task on MP-20, leaving generative performance comparison for the future work. The results are presented in Figure13, hyperparameters in Figure14.

*Table 13.* Performance of WyFormer with different representation. The values are slightly different from Figure3, as there we have tuned hyperparameters.

| Representation | Energy MAE, meV | Band Gap MAE, meV |
|---|---|---|
| Site symmetry only | 31.7 | 247.8 |
| Wyckoff letter | 30.5 | 234.0 |
| Site symmetry & *Enumeration* | 30.7 | 244.1 |
| Site symmetry & Harmonics | 29.7 | 238.7 |

| Parameter | Value |
|---|---|
| Element embedding size | 16 |
| Wyckoff letter embedding size | 27 |
| Site symmetry embedding size | 16 |
| Site *enumerations* embedding size | 7 |
| Harmonic vector length | 12 |
| Batch size | 500 |
| Number of fully-connected layers | 3 |
| Number of attention heads | 4 |
| Dimension of feed-forward layers inside Encoder | 128 |
| Dropout inside Encoder | 0.2 |
| Number of Encoder layers | 3 |

*Table 14.* Hyperparameters used in the ablation study.

## P. Sampling harmonic-encoded WPs

WP harmonic representation is a real-valued vector. But for each space group it can only take up to 8 possible values, so learning the full distribution of such vectors is not necessary. We tried the following procedure:

1. Take the harmonic representations of all the WPs in all space group

2. Use K-means clustering to find 8 cluster centers.

3. Separately for each space group, assign harmonic labels to each *enumeration*:

   (a) Compute the Euclidean distances between all cluster centers and all WPs in the SG

   (b) Choose the smallest distance. Assign the WP to the corresponding cluster, remove WP and the cluster center from consideration.

   (c) Repeat until all WPs are assigned

This way all we obtain a discrete prediction target with one-to-one mapping with *enumerations*, but where physically-similar values are grouped together. Experimentally, however, this modification reduces performance. When predicting spherical harmonics clusters, S.U.N. based on 1k CHGNet-relaxed structures was 34.0% as compared to 36.6% for enumerations-based model; S.U.N. based on 105 DFT structures S.U.N. was 19.1% vs 22.2%.

## Q. Superconductor critical temperature prediction

We used WyFormer to predict the critical temperature in superconductors on the 3DSC dataset (Sommer et al., 2023); obtained test MLSE of 0.81

## R. Token analysis

### R.1. WyFomer tokens

Tokens are formed from three parts: `(element, site symmetry, enumeration)`, for example: `(O, .m., 0)`. Considering all choices of space group Euclidean normalizer, there are 10904 unique tokens in MP-20. The distribution for MP-20 is present in Figure 13; for MPTS-52 in Figure 14.

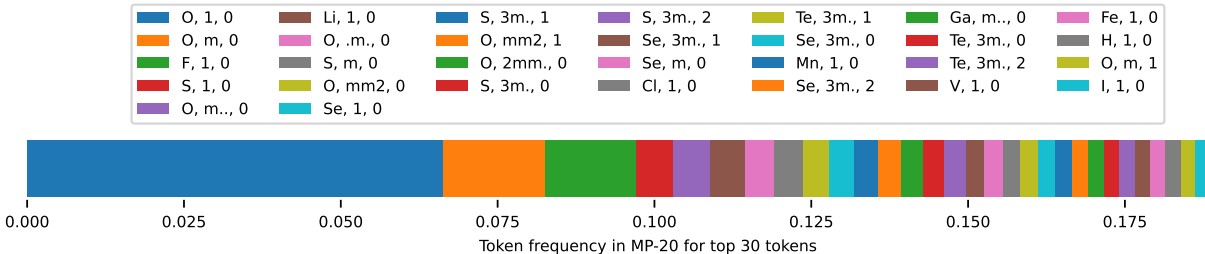

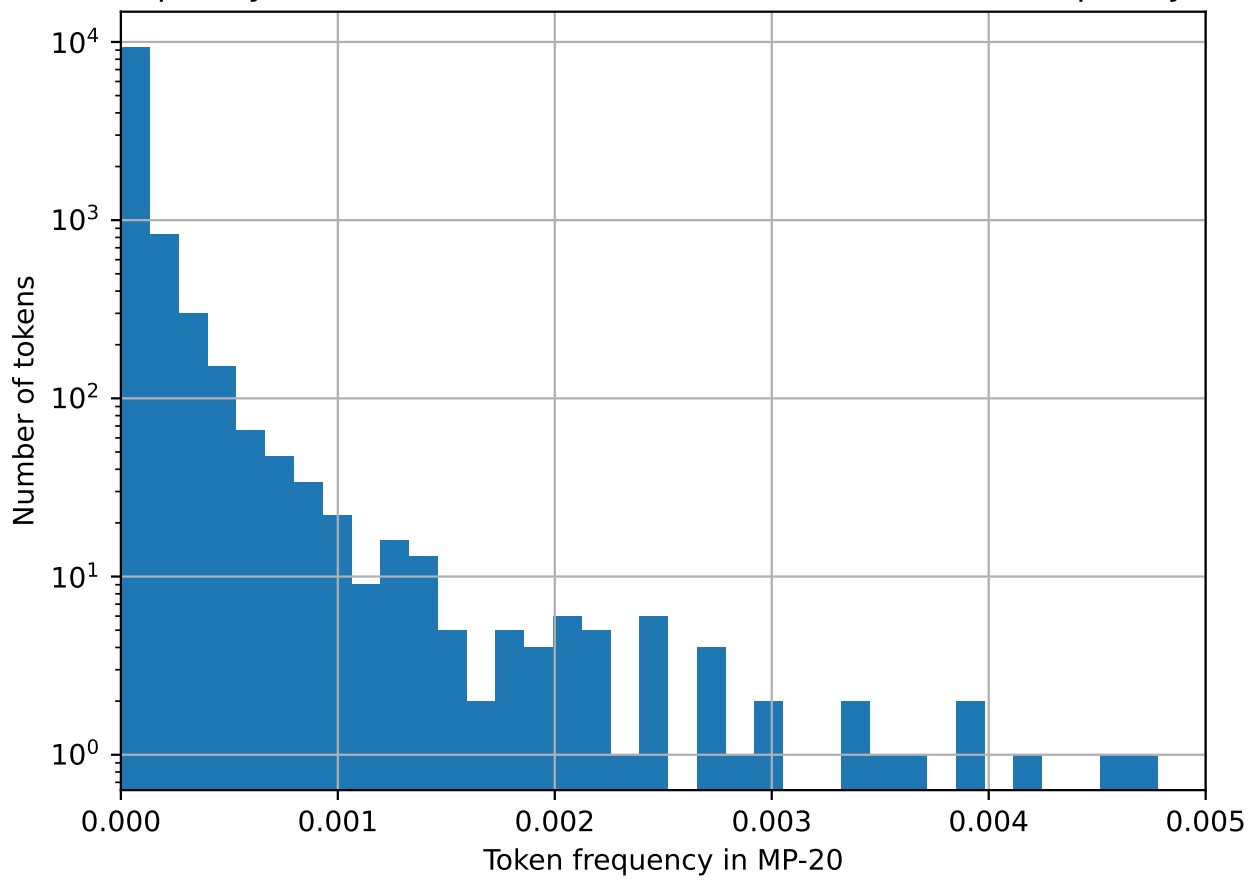

*Figure 13.* Distribution of tokens in MP-20

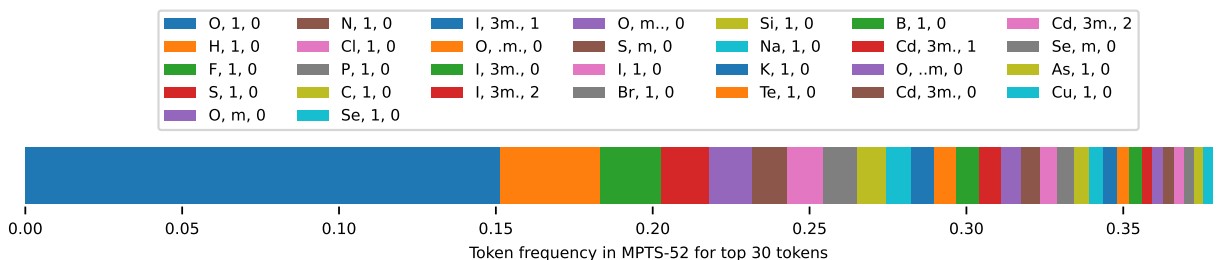

Token frequency in MPTS-52 for top 30 tokens

Token frequency distribution in MPTS-52 for 11066 tokens with frequency < (

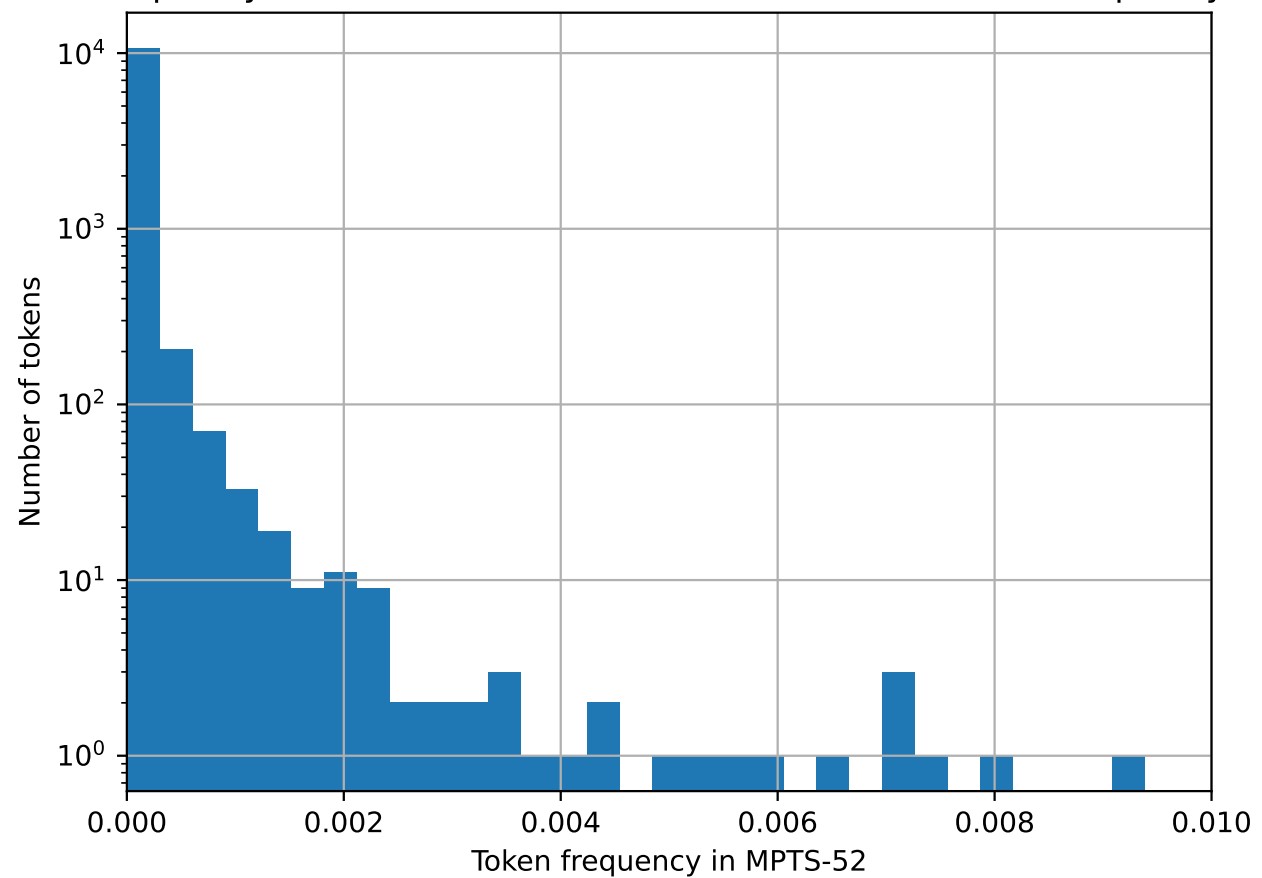

*Figure 14.* Distribution of tokens in MPTS-52

### R.2. Template tokens

In this section, we consider a different token structure (`space group number,` `site symmetry,` `enumeration`), which we will call template token. It does not correspond to token structure inside WyFormer, but the analysis of such tokens is interesting from the template novelty point of view. Considering all choices of space group Euclidean normalizer, there are 1047 unique template tokens in MP-20. The distribution is plotted in Figure 15. Wyckoff Transformer generates templates tokens not present in the training and validation datasets. For sample size of 9046 it produced 20 new template tokens; for comparison, the similarly-sized test dataset contains 21 new template tokens.

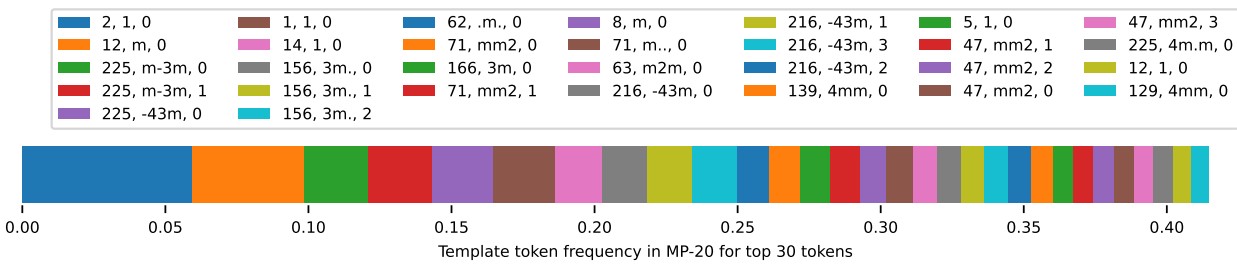

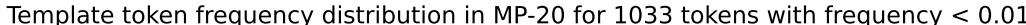

Template token frequency in MP-20 for top 30 tokens

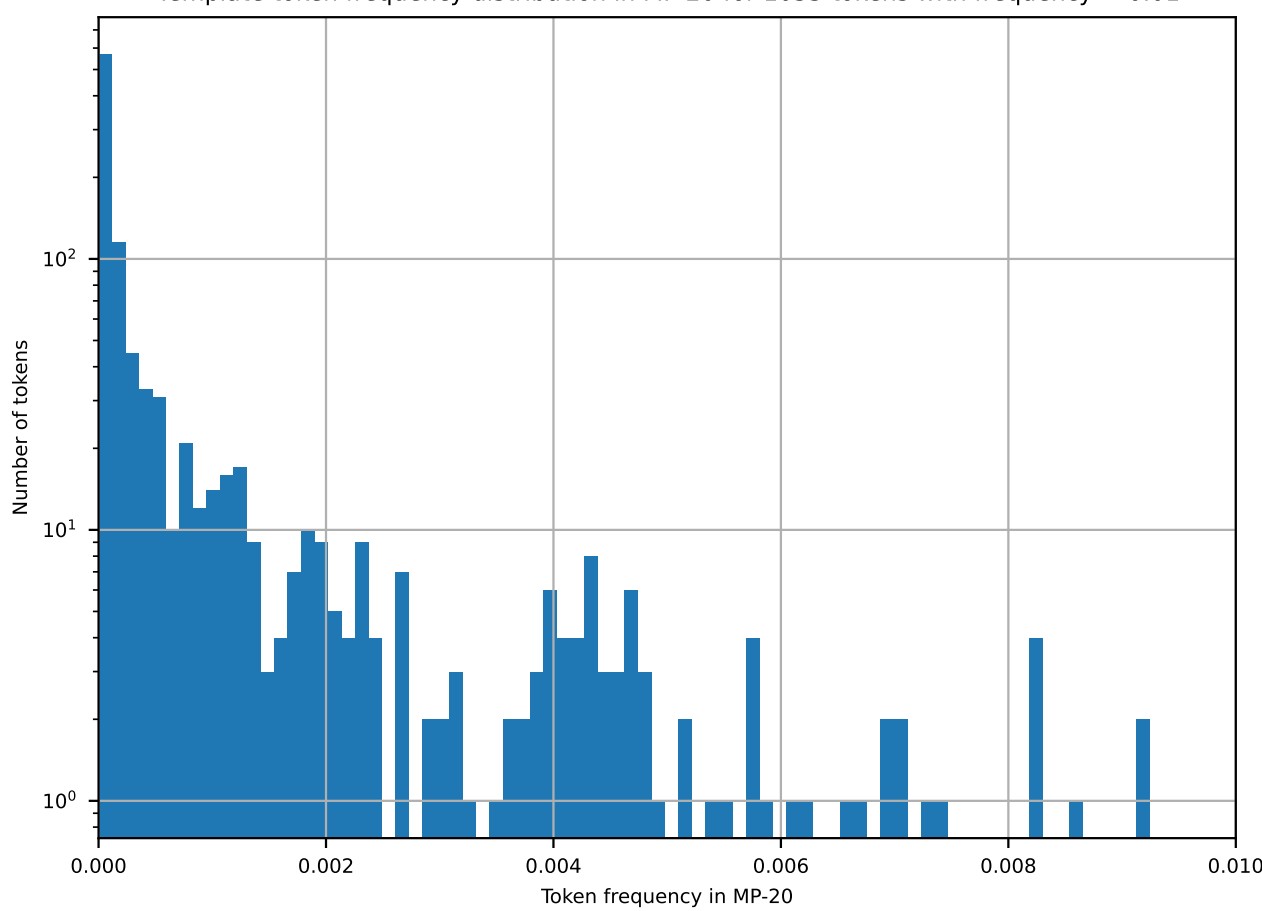

*Figure 15.* Distribution of template tokens in MP-20

# S. Comparison of enumerations for full Space groups

Figure 16.

**Wyckoff Positions: Groups 3 & 84 (P4₂/m) with Enumeration**

| Wyckoff Letter | Group 3 (P2) | | | | Group 84 (P4₂/m) | | | |
|---|---|---|---|---|---|---|---|---|
| | Mult. | Site Sym. | Enum. | Coordinates | Mult. | Site Sym. | Enum. | Coordinates |
| a | 1 | 2 | 0 | (0,y,0) | 2 | 2/m.. | 0 | (0,0,0), (0,0,1/2) |
| b | 1 | 2 | 1 | (0,y,1/2) | 2 | 2/m.. | 1 | (1/2,1/2,0), (1/2,1/2,1/2) |
| c | 1 | 2 | 2 | (1/2,y,0) | 2 | 2/m.. | 2 | (0,1/2,0), (1/2,0,1/2) |
| d | 1 | 2 | 3 | (1/2,y,1/2) | 2 | 2/m.. | 3 | (0,1/2,1/2), (1/2,0,0) |
| e | 2 | 1 | 0 | (x,y,z), (-x,y,-z) | 2 | -4.. | 0 | (0,0,1/4), (0,0,3/4) |
| f | - | - | - | - | 2 | -4.. | 1 | (1/2,1/2,1/4), (1/2,1/2,3/4) |
| g | - | - | - | - | 4 | 2.. | 0 | (0,0,z), (0,0,z+1/2), (0,0,-z), (0,0,-z+1/2) |
| h | - | - | - | - | 4 | 2.. | 1 | (1/2,1/2,z), (1/2,1/2,z+1/2), (1/2,1/2,-z), (1/2,1/2,-z+1/2) |
| i | - | - | - | - | 4 | 2.. | 2 | (0,1/2,z), (1/2,0,z+1/2), (0,1/2,-z), (1/2,0,-z+1/2) |
| j | - | - | - | - | 4 | m.. | 0 | (x,y,0), (-x,-y,0), (-y,x,1/2), (y,-x,1/2) |
| k | - | - | - | - | 8 | 1 | 0 | (x,y,z), (-x,-y,z), (-y,x,z+1/2), (y,-x,z+1/2), (-x,-y,-z), (x,y,-z), (y,-x,-z+1/2), (-y,x,-z+1/2) |

*Figure 16.* Different WPs can have a common site symmetry. In this case, they differ in coordinates. The corresponding column indicates the triples of coordinates of all the included atoms, where x, y, and z are the unfixed parameters that change from 0 to 1. Such collisions could be resolved using letters. However, as seen in the table, letters are not connected to symmetries and differ significantly between space groups. Therefore, we use an approach that numbers positions within a group of WPs with the same site symmetry. The ordering is performed in accordance with (Aroyo et al., 2006).

