# OpenReview forum: "Wyckoff Transformer: Generation of Symmetric Crystals"
_ICML.cc/2025/Conference — ICML 2025 poster_

### Official Review · Reviewer_QeGe · 2025-02-17

**Overall Recommendation:** 3

**Summary:**

This paper proposes WyFormer, a novel generative model for materials design that leverages Wyckoff positions to encode space group symmetry. The authors argue that symmetry rules are crucial for determining material properties, and that traditional material discovery approaches are limited by the vastness of the possible combinations of atoms. WyFormer aims to address this by generating stable materials with desired symmetries and properties using a permutation-invariant autoregressive model based on the Transformer architecture.

**Claims And Evidence:**

Yes

**Essential References Not Discussed:**

some missing refs about the transformer framework for CSP task:
1. Jin, Luozhijie, et al. "Transformer-generated atomic embeddings to enhance prediction accuracy of crystal properties with machine learning." Nature Communications 16.1 (2025): 1210.
2. Lin P, Chen P, Jiao R, et al. Equivariant Diffusion for Crystal Structure Prediction[C]//Forty-first International Conference on Machine Learning.
3. Yan, Keqiang, et al. "Periodic graph transformers for crystal material property prediction." Advances in Neural Information Processing Systems 35 (2022): 15066-15080.
4. Yan, Keqiang, et al. "Complete and efficient graph transformers for crystal material property prediction." arXiv preprint arXiv:2403.11857 (2024).

**Experimental Designs Or Analyses:**

Yes

**Methods And Evaluation Criteria:**

Yes

**Other Comments Or Suggestions:**

NA

**Other Strengths And Weaknesses:**

Strengths:

Novelty of approach: WyFormer’s focus on symmetry and its efficient tokenization approach are unique and innovative.
Comprehensive evaluation: The authors provide a thorough evaluation of WyFormer’s performance using various metrics and datasets.
Explainability: The use of Wyckoff positions and their encoding provides a transparent and interpretable representation of material structures.

Weaknesses:

Limited scalability: The current implementation of WyFormer is limited to structures with a maximum of 20 atoms per unit cell. Scalability to larger structures is a challenge that needs to be addressed in future work.
Computational cost: The inference process for WyFormer can be computationally expensive, especially for larger structures. Optimizations and parallelization techniques are needed to improve efficiency.
Comparison with other Wyckoff-based models: While the paper compares WyFormer with a few existing models, a more comprehensive comparison with other Wyckoff-based models would be beneficial.

**Questions For Authors:**

1. How does WyFormer handle cases where multiple Wyckoff positions are possible for a given site symmetry and enumeration?
2. What are the limitations of the spherical harmonics-based enumeration representation?
3. How does WyFormer handle materials with mixed-valence properties?
4. Are there any plans to extend WyFormer to handle larger structures?
5. How can the computational cost of WyFormer be reduced?

**Relation To Broader Scientific Literature:**

WyFormer builds upon the existing literature on crystal structure generation and material property prediction.

**Theoretical Claims:**

Yes

---

> ### Author Rebuttal · Authors · 2025-03-31
>
> We sincerely thank the reviewer for the insightful feedback. We appreciate their recognition of the **novelty of WyFormer's symmetry–focused and efficient tokenization approach**, **comprehensive evaluation across various metrics and datasets**, and the **explainability** provided by the use of Wyckoff positions.
>
> We will add the proposed references.
>
> **Larger structures**
>
> We have trained WyFormer on MPTS-52, which contains up to 52 atoms per unit cell, without any issues. The values of the metrics are present in the last rows of Tables 1 and 2. As expected, on MPTS-52 WyFormer shows higher novelty and template novelty. We estimated S.S.U.N with CHGNet: 24.4% on MPTS-52, vs 35.2% on MP-20. This reflects the increased difficulty, and shows that WyFormer is still very much capable of generating stable structures in this setting.
>
> **We are glad you have asked about computational speed!**
>
> WyFormer **has the fastest inference** of all the models. Inference speed evaluation results are present in Appendix F Table 6; we did some more experiments is addition to it, will add them to the camera-ready version:
>
> Generating a batch of $10^5$ Wyckoff representations takes 25 seconds, of which 5 seconds are spent generating pytorch tensors, and 20 seconds on decoding them into Python dictionaries containing Wyckoff representations. The latter part hasn’t been optimized. In total, generation takes **0.05 GPU ms and 4.8 CPU ms per structure**.
>
> Obtaining unrelaxed structures using pyXtal takes 100 CPU ms / structure.
>
> Relaxing the structure is the most expensive step:
>
> 1. DiffCSP++ takes 14 minutes to produce 1000 structures at 840 GPU ms / structure. Note that we modified the code to remove the inference of atom types, so it runs faster compared to the original version.
> 2. CHGNet: 112 GPU s / structure for MP-20 on NVIDIA A40
>
> **Baselines**
>
> DiffCSP: the authors don’t report speed. On our machine, generating 10000 structures on GPU took 1 hour, at **360 GPU ms per structure**.
>
> DiffCSP++: the authors don’t report speed. On our machine, generating 27135 structures took 6 hours, at **1.25 CPU+GPU seconds per structure**
>
> CrystalFormer paper: “It takes 520 seconds to generate a batch size 13,000 crystal samples on a single A100 GPU”, which translates to a generation speed of **40 ms per sample**.
>
> FlowMM: The authors also do not publish inference time or model weights. They claim to be 3x faster than DiffCSP in terms of integration steps.
>
> WyCryst paper: “Latent space sampling 1 CPU second/2000 structures; PyXtal generation 2 CPU core seconds/structure”
>
> > Comparison with other Wyckoff-based models: While the paper compares WyFormer with a few existing models, a more comprehensive comparison with other Wyckoff-based models would be beneficial.
>
> WyCryst, CrystalFormer, and DiffCSP++ are Wyckoff-based models, and are included in our baselines. In response to CSRK, we have also made a comparison with a concurrent work, SymmCD:
>
> 1. WyCryst underperforms on all the metrics
> 2. DiffCSP++ has lower stability, and as we show in appendix K, the lack of template novelty limits the diversity
> 3. CrystalFormer has low novelty – a sign of overfitting. It also produces a sizable fraction of a priori structurally invalid crystals.
> 4. SymmCD has similar stability, but lower template novelty
>
> > How does WyFormer handle cases where multiple Wyckoff positions are possible for a given site symmetry and enumeration?
>
> 1. In terms of group theory, the combination of space group, site symmetry and enumeration uniquely defines a WP. This is essentially the definition of *enumerations*, see the [figure](https://www.notion.so/Enumerations-1c775a35da3680efb760f6dcb7c03ab1?pvs=21) for more details.
> 2. It is possible for different crystallographic orbits to have the same WP; if the WP contains a free parameter, the atoms can still occupy different locations in 3D space. WyFormer handles it naturally by repeating the site symmetry and enumeration.
>
> > What are the limitations of the spherical harmonics-based enumeration representation?
>
> They are not invertible and can’t be directly used for structure generation. We have implemented the clustering algorithm from Appendix P, results pending.
>
> > How does WyFormer handle materials with mixed-valence properties?
>
> WyFormer handles materials with mixed-valence properties implicitly, rather than explicitly. Its tokenization scheme and symmetry-aware representation, which explicitly include space group symmetry and site symmetry, allow it to generate and learn from materials where atoms occupy non-equivalent crystallographic sites with differing local environments, often the cause of mixed valence, without directly specifying valence states
>
> Adding explicit oxidation states should be possible as an additional feature in WyFormer, a direction for future research. Thank you for the idea!

---

### Official Review · Reviewer_CSRK · 2025-03-06

**Overall Recommendation:** 3

**Summary:**

The author uses Wyckoff positions as the basis for structure representation and develops a permutation-invariant autoregressive model based on the Transformer architecture, with the absence of positional encoding.

## update after rebuttal

My concern has been addressed.

**Claims And Evidence:**

Claims made in the submission are supported by clear and convincing evidence.

**Essential References Not Discussed:**

NA

**Experimental Designs Or Analyses:**

I have reviewed the experimental designs, and they are reasonable.

**Methods And Evaluation Criteria:**

Proposed methods and evaluation criteria make sense for the problem or application at hand.

**Other Comments Or Suggestions:**

1. To facilitate reading, you could use mathematical formulas in section 2 to describe the problem your method aims to solve, as well as the method itself, rather than only describing it in text.

2. You may consider adding a flowchart of the proposed method in section 2. If your Figure 6 is a flowchart, it would be better to move it to section 2.

**Other Strengths And Weaknesses:**

Strengths:

Empirically, the proposed model outperforms baseline methods in generating novel, symmetric, diverse materials conditioned on space group symmetry.


Weaknesses:

The writing of the paper is quite poor. In section 2, the problem and method to be solved are not described using mathematical formulas but only explained with text. Additionally, section 2 lacks a flowchart of the proposed method. This makes it difficult for readers to follow and understand the content.

**Questions For Authors:**

1. In the experimental section, methods WyFormerRaw and WyForDiffCSP++ are mentioned. What is the difference between these two methods, and why are they not described in section 2? Are they related to DiffCSP++ mentioned in section 2.4?

2. Does WyFormer only generate Wyckoff positions? How are the lattice matrix-related parameters generated? According to the description in section 2.4, "Structure generation," is WyFormer used to generate the Wyckoff representation first, and then another method is used to generate the lattice matrix-related parameters? If so, what advantages does WyFormer have compared to methods like DiffCSP++, which can generate the entire structure in one step while ensuring the accuracy of the Wyckoff positions? How does WyFormer compare to SymmCD [1], which also utilizes Wyckoff positions?

[1]Levy, Daniel, et al. "SymmCD: Symmetry-Preserving Crystal Generation with Diffusion Models." AI for Accelerated Materials Design-NeurIPS 2024.

3. Based on your description in the main text and your title, your focus seems to be on generation methods. Why, then, is your method also applied to prediction tasks? Moreover, your prediction method shows limited effectiveness, and the comparison methods in Table 3 are not the latest, as they do not include the baseline proposed in 2024.

4. The performance of WyFormer-related methods in Table 2 does not seem to be particularly outstanding.

**Relation To Broader Scientific Literature:**

NA

**Theoretical Claims:**

Proofs for theoretical claims are correct.

---

> ### Author Rebuttal · Authors · 2025-03-31
>
> Thank you for your thorough and constructive review of our submission. We greatly appreciate the time and effort you dedicated to evaluating our work. We are particularly encouraged by your positive feedback on the key aspects of our paper: **reasonable experimental results leading to  clear and convincing evidence that WyFormer outperforms baseline methods in generating novel, symmetric, diverse materials conditioned on space group symmetry**.
>
> Regarding the weaknesses you pointed out, we sincerely regret that, according to the ICML guidelines, we are unable to update the submitted PDF at this stage. However, we want to assure you that we have taken your feedback to heart and will include the suggested improvements into the camera-ready version:
>
> 1. We will **introduce mathematical formulas and [pseudocode](https://wyckoff.notion.site/Pseudocode-1c875a35da36808289a8fcd3e43b56dc) to describe the problem our method aims to solve and the method itself**
> 2. [Updated flowcharts](https://www.notion.so/WyFormer-Flowcharts-1c775a35da368002a4caf977dd980296)
>
> >Does WyFormer only generate Wyckoff positions? …
>
> Wyckoff positions and chemical elements
> >what advantages does WyFormer have compared to methods like DiffCSP++, which can generate the entire structure in one step while ensuring the accuracy of the Wyckoff positions? How does WyFormer compare to SymmCD, which also utilizes Wyckoff positions?
>
> **WyFormer just works better than 1–step methods & its inference speed is four orders of magnitude faster than that of diffusion models** (Appendix F). Similarly to the situation in LLMs, autoregressive generation provides a better inductive bias for discrete Wyckoff positions, while diffusion provides a better one for continuous coordinates
> 1. DiffCSP++ *requires as input* a template with Wyckoff positions. WyFormer generates those templates. In the DifffCSP++ paper the authors use templates from the training dataset. As we show in Appendix K, the lack of template novelty limits the generated sample diversity. DiffCSP++ also has lower S.(S).U.N.: 7.6% vs 12.8%.
> 2. CrystalFormer has low novelty, which means that the model has been overfitted, and the structures are similar to the training dataset. It also produces a sizable fraction of structurally invalid crystals.
> 3. WyCryst suffers from abyssal novelty, stability and distribution similarity metrics.
> 4. SymmCD is a [concurrent work](https://icml.cc/Conferences/2025/ReviewerInstructions), and was not included in the experiments in the original submission. We have done them and will include in the camera–ready version:
>
> |Method|Novel Uniques Templates (#) ↑|P1 (%) ref = 1.7|Space Group χ2 ↓|DFT S.(S).U.N. (%) ↑|CHGNet S.(S).U.N. (%) ↑|
> |-|-|-|-|-|-|
> |SymmCD |161|2.35|**0.24**|**12.1 (12.1)**|**34.1 (33.2)**|
> |WyForDiffCSP++ | **186**|**1.46**|**0.21**|**12.7 (12.7)**|**36.6 (35.9)**|
>
> WyFormer achieves higher number of novel uniques templates; higher S.(S).U.N., but this difference is not statistically significant.
> >Based on your description in the main text and your title, your focus seems to be on generation methods. Why, then, is your method also applied to prediction tasks? Moreover, your prediction method shows limited effectiveness, and the comparison methods in Table 3 are not the latest, as they do not include the baseline proposed in 2024.
>
> We show that it is possible to do reasonable property prediction in the Wyckoff space, laying groundwork for property-conditioned generation.
>
> Predicting properties also serves to support our core assumption: crystal symmetries play a crucial role in the properties of matter, including the ones not included in MP–20, but crucial for material design. We don’t intend for a model using only symmetries to outperform models using whole structures (although we sometimes see this), just prove that symmetry information alone largely (but not completely) determines properties of the material.
> > In the experimental section, methods WyFormerRaw and WyForDiffCSP++ are mentioned. What is the difference between these two methods? …
>
> Apologies for the confusion, we’ll add the definitions prominently to the camera-ready version. Different suffixes denote different ways to obtain the final structure from the Wyckoff representation:
> 1. WyFormer uses CrySPR and CHGNet
> 2. WyForDiffCSP++ uses DiffCSP++
> 3. WyFormerRaw samples an unrelaxed structure with pyXtal
>
> > The performance of WyFormer–related methods in Table 2 does not seem to be particularly outstanding.
>
> *None of the methods are outstanding in Table 2*, because the **metrics in Table 2, except novelty, were proposed in [2021](https://arxiv.org/abs/2110.06197) and are mostly saturated,** the most important metrics are in Table 1.
>
> 1. WyFormer is **within 1% of the best value in 4 out of 5** %-based metrics: Novelty, Structural Validity, COV-R and COV-P.
> 2. In terms of EMD, out of 6, WyFormer ranks 3rd for $ρ$, tied 3rd–4th for $E$, and 1st for $N_\text{elem}$

---

> > ### Comment · Reviewer_CSRK · 2025-04-08
> >
> > Thank you for your reply. I have understood some details. I will increase the score to 3, but I still recommend reorganizing the content of the paper when the PDF can be modified to reduce the burden on readers, especially in the descriptions of the methods.

---

### Official Review · Reviewer_PsvD · 2025-03-14

**Overall Recommendation:** 4

**Summary:**

The paper proposes a transformer-based model called WyFormer to learn the representation of crystal structures considering their symmetry information. The model represents crystals as discrete tokens encoding space group, chemical elements, and Wyckoff positions rather than using 3D coordinates. Through their experiments, the authors demonstrate that WyFormer can generate novel materials with proper crystal symmetry, outperforming baselines on symmetry metrics while maintaining competitive performance in terms of diversity, efficiency and other metrics. Additionally, the Wyckoff representation can complement other crystal generation methods.

## Update after rebuttal
I believe the authors have presented convincing evidence and analysis to support their experimental results. With the complementary results added to the draft, it should be a solid contribution to the field.

**Claims And Evidence:**

The main claim is that WyFormer generates crystal structures with better symmetry properties than baseline methods while maintaining competitive stability. I believe this claim is supported by the experiments.

**Essential References Not Discussed:**

N/A

**Experimental Designs Or Analyses:**

Experiments are reasonably designed.

**Methods And Evaluation Criteria:**

Yes. The authors evaluate the generated structures on metrics across elemental distribution, stability and space group distribution.

**Other Comments Or Suggestions:**

1. I suggest clear definitions for some abbreviations before referencing, for example WyFormerDiffCSP++ and WyLLM.

**Other Strengths And Weaknesses:**

Strengths:
1. The focus on symmetry addresses the limitation in existing generative models like CDVAE, DiffCSP and FlowMM.
2. The authors provide supplementary experiments on fine-tuning an LLM like CrystalLLM, which offers some interesting insights.
3. The experimental results with WyFormerDiffCSP++ suggest a future direction of combining discrete symmetry modeling with continuous coordinate refinement.

Weakness:
1. The formulation and machine learning techniques explored in this approach are relatively simple, with limited technical innovation from a machine learning perspective, offering modest insights for broader ML researchers.

**Questions For Authors:**

1. How does WyFormer's performance scale with larger sample size, as currently very limited structures are sampled and reported in Table 1 and even fewer structures are sent for DFT evaluation?
2. Have you analyzed the discrepancy between crystal structures before and after relaxation? How much does the performance rely on relaxation?
3. The S.U.N. metric reported across methods is relatively low (<20%), which may not fully capture the uniqueness and diversity of crystal structures. Have you explored alternative evaluation metrics?
4. You mentioned selecting one sample with the lowest energy out of 6 random initializations. What variance in energy did you observe across these initializations?
5. How are structures generated with WyLLM variations being relaxed? What considerations behind not adopting M3GNet used in the CrystalTextLLM paper? Some additional thoughts, have you observed differences in decomposition energy when using different relaxation methods, and what are the key tradeoffs between WyFormer's specialized architecture and fine-tuned LLMs based on your comparative experiments?

**Relation To Broader Scientific Literature:**

The paper demonstrates the integration of Wyckoff representation with diffusion models as a future direction for crystal structure generation.

**Theoretical Claims:**

The claims are valid.

---

> ### Author Rebuttal · Authors · 2025-03-31
>
> We appreciate your positive feedback and recognition of our main claim that **WyFormer addresses  the limitation in existing generative models and achieves better symmetry in generated crystals with competitive stability**, and its support by **reasonably designed experiments**. We are especially encouraged by the recognition of integration of Wyckoff representation with diffusion models as a promising research direction for crystal structure generation.
>
> We acknowledge the simplicity of ML techniques. The core concept—combining autoregressive generation of a discrete high-level representation with diffusion-based refinement—may hold potential, if modest, interest for the wider ML community, possibly extending to areas like music/video generation and robotic motion planning. Simplicity also facilitates easier understanding and implementation, quoting an ICML–[endorsed](https://icml.cc/Conferences/2025/ReviewerInstructions) [meme](https://researchonresearch.blog/wp-content/uploads/2024/10/slide9.jpeg).
>
> > How does WyFormer's performance scale with larger sample size, as currently very limited structures are sampled and reported in Table 1 and even fewer structures are sent for DFT evaluation?
>
> Each sampled structure is sampled independently, so the performance per se doesn’t change with sample size – **except for uniqueness.** In Appendix K we present the number of unique structures as a function of the sample size, highlighting the ability of WyFormer to generate a diverse set of structures.
>
> > Have you analyzed the discrepancy between crystal structures before and after relaxation? How much does the performance rely on relaxation?
>
> - For all symmetry–based methods the space group is the same before and after DFT for > 90% examples; 44% for FlowMM and 55% for DiffCSP
> - Figure 9 presents root mean squared deviation (RMSD) of DFT-unrelaxed structures from DFT-relaxed; for > 90% of examples RMSD < 0.2 Å
> - Table 7 presents the very similar proxy metric values for generated structures with and without CHGNet relaxation
>
> In conclusion, the materials before and after relaxation are similar, and relaxation meaningfully affects performance for about 10% of examples.
>
> > The S.U.N. metric reported across methods is relatively low (<20%), which may not fully capture the uniqueness and diversity of crystal structures. Have you explored alternative evaluation metrics?
>
> Good question! Firstly low S.U.N. value is actually good in the sense that this metric is not saturated, as opposed to the ones proposed by [Xie et. al in 2021](https://arxiv.org/abs/2110.06197). We additionally explore the uniqueness and diversity in two ways:
>
> 1. By counting novel unique templates. This captures physically meaningful sample diversity: if two materials have different templates, their physical properties will definitely be different, while two structures which don’t match precisely can be similar.
> 2. In Appendix K we present an evaluation on uniqueness of WyFormer-generated structures as a function of sample size.
>
> > You mentioned selecting one sample with the lowest energy out of 6 random initializations. What variance in energy did you observe across these initializations?
>
> The mean std is 0.14 eV/atom, see the [figure](https://www.notion.so/CHGNet-energy-standard-deviation-1c775a35da3680fbac4de4cdaa1a3e40?pvs=21) for the full distribution. We’ll add it to the camera–ready version.
>
> > How are structures generated with WyLLM variations being relaxed? What considerations behind not adopting M3GNet used in the CrystalTextLLM paper?
>
> Initially we used M3GNet, but then switched to newer CHGNet. However, for WyLLM experiments we used DiffCSP++, as it showed better performance compared to CHGNet-based structure generation, see WyFormer vs WyForDiffCSP++ in tables 1 and 2.
>
> > have you observed differences in decomposition energy when using different relaxation methods
>
> We used two relaxation and energy estimation methods: CHGNet and DFT. CHGNet inflates S.U.N., for example see the table below. Pearson correlation between structures’ stability determined by DFT and CHGNet was in range 0.3–0.4, so CHGNet is still useful.
>
> > what are the key tradeoffs between WyFormer's specialized architecture and fine-tuned LLMs based on your comparative experiments?
>
> The main one is, of course, the computational cost: WyFormer has 150k parameters to gpt-4o-mini-2024-07-18 8B; WyFormer’s inference time is at least an order of magnitude faster. The comparison is also unfair as the LLM has seen much more data than WyFormer. *Despite all of these advantages, there is no clear gain from using an LLM.* In addition to the proxy metrics reported in the paper, we have computed DFT & CHGNet relaxation:
>
> |  | DFT S.(S).U.N. (%)↑ | CHGNet S.(S).U.N. (%) ↑ |
> | --- | --- | --- |
> | WyLLM-naive-DiffCSP++ | 9.0 (9.0) | 31.6 (30.9) |
> | WyForDiffCSP++ | **12.8 (12.8)** | **36.6 (35.9)** |
>
> We’ll update the camera–ready version to include these results.

---

### Official Review · Reviewer_YPYZ · 2025-03-14

**Overall Recommendation:** 2

**Summary:**

This paper introduces WyFormer, a Transformer-based architecture to generate Wyckoff sites of crystal structures. The key idea is the tokenization approach to convert the structure into a sequence of the space group and Wyckoff sites, and a permutation-invariant auto-regressive transformer for sequence generation. The proposed model can further generate the entire structures with PyXtal-based initializations and refinement via CHGNet or DiffCSP++. Results on MP-20 showcases the effectiveness of this method.

**Claims And Evidence:**

The claims in this paper are supported by clear evidence.

**Essential References Not Discussed:**

Most of the essential references in this field are already be discussed.

**Experimental Designs Or Analyses:**

The experiments in this work are well-designed. Notably, Table 1 introduces symmetry-based metrics to evaluate the model’s symmetry-aware generation performance.

**Methods And Evaluation Criteria:**

The main part of the proposed method is described vaguely. The reviewer finds that the concept of "enumerations" plays a crucial role in the tokenization process. Unlike the traditional encoding of Wyckoff positions using letters and multiplicities, the proposed method employs "enumerations." The authors are suggested to provide a clearer explanation of the differences between these representations and the advantages of the enumeration-based approach.

**Other Comments Or Suggestions:**

Some typos or confusing sentences could be found:

Line 179 (Left): is it -> it is

Line 247 (Left): fist -> first

Line 165-166 (Right): "The reason is that in multihead attention, different heads look at continuous blocks of the input vector." The meaning of this sentence is unclear. Could the authors clarify what is meant here?

**Other Strengths And Weaknesses:**

1. The allocation of content in this paper is a key issue. The introduction spends a significant amount of space discussing space groups and Wyckoff positions, while the method section (Section 2) is very brief, which hinders the reader's understanding of the proposed method. The reviewer suggests that the authors consider restructuring the introduction by moving the theoretical background and data representation into a preliminary section, or relocating some of the redundant content to the appendix. This would free up more space to provide a more detailed explanation of the method (for instance, the content in Appendix C would be more appropriate for the main body of the paper). The reviewer would be willing to improve the score if the authors enhance the readability of the paper.

2. Figure 4 is somewhat ambiguous and may lead readers to mistakenly believe that element, site symmetry, and enumeration are integrated as a single input-output token. However, as described in lines 229-240, these three aspects are auto-regressive outputs. The reviewer suggests that the authors revise Figure 4 to reduce this ambiguity.

**Questions For Authors:**

1. Could the authors give more experimental evidence of the advantages of using the site symmetries and enumerations instead of Wyckoff letters and multiplicities?

**Relation To Broader Scientific Literature:**

Generating high-symmetry crystal structures is an important topic. Recently, DiffCSP++ [A] proposed a generation framework that conditions on space groups and Wyckoff sites. The proposed framework extends this approach by generating these conditions directly and introducing a two-stage generation process to sample high-symmetry crystal structures from scratch.

[A] Jiao, Rui, et al. "Space group constrained crystal generation." ICLR 2024.

**Theoretical Claims:**

The theoretical definitions and claims in this paper are correct.

---

> ### Author Rebuttal · Authors · 2025-03-31
>
> Thank you for your insightful review and for recognizing the strengths of our paper, particularly the **importance of generating high–symmetry crystal structures,** the **well–designed experiments, and the clarity of evidence** they produce!
>
> We appreciate your detailed comments aimed at bettering the clarity and presentation of our work, and the willingness to improve the score upon enchantment the paper readability. As per the ICML guidelines, we are unable to update the submitted PDF at this stage. However, we want to assure you that we have implemented the proposed changes to enhance the readability and clarity of our paper. **The following represents the text of the key changes that will be incorporated into the camera-ready version:**
>
> 1. **Enumerations**. Different WPs can share the site symmetry. *Enumerations* allow us to disambiguate them by enumerating such WPs from 0 up to 6 in the conventional order from [ITA](https://www.iucr.org/news/newsletter/volume-25/number-1/it-vol-a-6th-ed) (one that’s used for assigning letters). Enumeration is separate for each site symmetry. **[Here](https://www.notion.so/Enumerations-1c775a35da3680efb760f6dcb7c03ab1?pvs=21) we have prepared a figure illustrating the concept.** The key advantage of using site symmetry plus *enumeration* over Wyckoff letters is that they allow us to compartmentalise the part of the token whose definition depends on the space group into the *enumeration,* leaving the universally-defined site symmetries to be learned by the model.
> 2. **Content allocation.** We will restructure the introduction to be more concise and move some of the theoretical background on space groups and Wyckoff positions, as well as the detailed data representation, into a dedicated Appendix as you suggested. The content currently in Appendix C on spherical harmonics will be moved to this revised Section 2 to provide a more detailed explanation of the method in the main body of the paper. We have also prepared a [flowchart](https://www.notion.so/WyFormer-Flowcharts-1c775a35da368002a4caf977dd980296?pvs=21) and [pseudocode](https://www.notion.so/Pseudocode-1c875a35da36808289a8fcd3e43b56dc?pvs=21) to further explain WyFormer.
> 3. **Figure 4** is corrected [here](https://www.notion.so/Figure-4-Tokens-1c775a35da368051b878c36c46c58d56?pvs=21).
> 4. **Typos** fixed
> 5. **Multihead attention.** The sentence "The reason is that in multihead attention, different heads look at continuous blocks of the input vector" refers to a potential limitation when applying multihead attention to the concatenated embeddings of different features (element, site symmetry, and enumeration).
>     1. Concatenating embeddings creates distinct, contiguous blocks of information in the input vector.
>     2. Standard multihead attention processes the input by splitting it into several independent heads. If these heads operate on continuous blocks corresponding to single feature types (e.g., only element embeddings), they might fail to learn relationships between different types of features (e.g., the relationship between an element and its site symmetry).
>     3. To address this, we apply a linear layer after concatenating the embeddings. This step helps to mix the features, ensuring that each attention head has access to information from all embedding types and can therefore learn cross-feature relationships.
>
> > Could the authors give more experimental evidence of the advantages of using the site symmetries and enumerations instead of Wyckoff letters and multiplicities?
>
> We have conducted an experiment by training a variant of WyFormer that uses Wyckoff letters instead of site symmetry + enumeration, with the same hyperparameters, and generating 1k examples. We then used DiffCSP++ to obtain the structures, and computed DFT for 105 structures. While using letters results in a higher number of novel unique templates, crucially, **site symmetry achieves 2x S.U.N. and S.S.U.N. compared to letters**. We’ll add these results to the camera-ready version.
>
> | Method/Metric | Novel Uniques Templates (#) ↑ | P1 (%) ref = 1.7 | Space Group χ2 ↓ | DFT S.(S).U.N. (%) ↑ |
> | --- | --- | --- | --- | --- |
> | WyFormer-letters-DiffCSP++ | **250** | 1.16 | **0.21** | 6.7 (6.7) |
> | WyFormer-DiffCSP++ | 186 | **1.45** | **0.21** | **12.8 (12.8)** |
>
> A small remark: multiplicity mentioned by the reviewer can be used with both letters and site symmetries. We have tried including it in preliminary experiments, which didn’t lead to improvement, so we don’t use it.

---

### Decision · Program_Chairs · 2025-05-01

**Decision:**

Accept (poster)

**Comment:**

This paper presents a method named WyFormer, which is designed to generate crystal structures autoregressively, by generating the space group of the unit cell, the elements and the Wyckoff positions. There is no consensus among reviewers about the overall assessment and recommendation of the paper. On the one hand, the method seems solid and the results are convincing though not groundbreaking. On the other hand, the structure, clarity and writing of the paper could be largely improved. This has been noted by several reviewers and is also my own assessment. For these reasons, I am recommending a shy acceptance. I think the technical quality of the method deserves acceptance, but the presentation might not meet the minimum standards.